# Cooperative Visual-SLAM System for UAV-Based Target Tracking in GPS-Denied Environments: A Target-Centric Approach

**Juan-Carlos Trujillo** [1][iD]**, Rodrigo Munguia** [1,*][iD]**, Sarquis Urzua** [1] **and Antoni Grau** [2][iD]

[1] Department of Computer Science, CUCEI, University of Guadalajara, Guadalajara 44430, Mexico;
juancarlos_max@hotmail.com (J.-C.T.); isi.sarquis@gmail.com (S.U.)

[2] Department of Automatic Control, Technical University of Catalonia UPC, 08034 Barcelona, Spain;
antoni.grau@upc.edu

[*] Correspondence: rodrigo.munguia@academicos.udg.mx

**Abstract:** Autonomous tracking of dynamic targets by the use of Unmanned Aerial Vehicles (UAVs) is a challenging problem that has practical applications in many scenarios. In this context, a fundamental aspect that must be addressed has to do with the position estimation of aerial robots and a target to control the flight formation. For non-cooperative targets, their position must be estimated using the on-board sensors. Moreover, for estimating the position of UAVs, global position information may not always be available (GPS-denied environments). This work presents a cooperative visual-based SLAM (Simultaneous Localization and Mapping) system that allows a team of aerial robots to autonomously follow a non-cooperative target moving freely in a GPS-denied environment. One of the contributions of this work is to propose and investigate the use of a target-centric SLAM configuration to solve the estimation problem that differs from the well-known World-centric and Robot-centric SLAM configurations. In this sense, the proposed approach is supported by theoretical results obtained from an extensive nonlinear observability analysis. Additionally, a control system is proposed for maintaining a stable UAV flight formation with respect to the target as well. In this case, the stability of control laws is proved using the Lyapunov theory. Employing an extensive set of computer simulations, the proposed system demonstrated potentially to outperform other related approaches.

**Keywords:** unmanned aerial vehicles; cooperative visual SLAM; state estimation; target tracking; observability; flight formation control

## 1. Introduction

Nowadays, Unmanned Aerial Vehicles (UAVs), computer vision techniques, and flight control systems have received great attention from the research community in robotics. This interest has as a consequence the development of systems with a high degree of autonomy. UAVs are very versatile platforms. In particular, aerial robots with rotary wings, as the quadcopters, allow great flexibility of movements, which makes them very useful for several tasks and applications [1,2]. Multi-robot systems have also received great attention from the robotics research community. This attention is motivated by the inherent versatility that those systems have to perform tasks that could present difficulties to be realized by a single robot. The use of several robots can have advantages such as an increase of robustness, better performance, and efficiency [3,4].

In this context, one important research problem is the control and coordination of a team of UAVs flying in formation with respect to a non-cooperative moving target. In this context, a fundamental task that must be addressed to control the flight formation is the estimation of the positions of UAVs and

the moving target. For most applications, GPS (Global Positioning System) still represents the main alternative for addressing the localization problem of UAVs. Nevertheless, the use of GPS presents some drawbacks, for instance, in scenarios where GPS signals are jammed intentionally [5] or when the precision error is substantial and they provide poor operability due to multipath propagation (natural and urban canyons [6,7]). In addition, there exist scenarios (e.g., indoor) where GPS is completely unavailable. Moreover, for non-cooperative targets, their position has to be estimated using on-board sensors.

In such scenarios, when considering a team of aerial robots flying in formation with respect to a particular target, the absolute pose (e.g., world-centric configuration) is not so important compared to the relative position information between the aerial vehicles and the target (e.g., Robot-centric configuration). In this case, sensors can help to estimate the required position information such as range sensors (laser or sonar) and radio-frequency (RF) tag-like-sensors (see [8–11]). However, these kinds of sensors are expensive and sometimes heavy, and their use in outdoor environments is limited somehow. Moreover, some of these sensors need the cooperation of the moving target. Active laser systems (e.g., LiDAR [12]) represent a very interesting sensing technology; they can operate under any visibility condition (i.e., both day and night, unlike cameras) and can directly provide 3D measurements about the surrounding environment. On the other hand, LiDAR is generally expensive overweighting the system for certain applications presenting moving parts which can induce some errors. Stereo systems [13] and depth cameras [14] also can obtain 3D information about the target and the environment; however, this kind of system requires that the objects to be measured are near the sensor, thus considerably limiting their range of application.

In the above context, more works have appeared focusing on the use of cameras to develop navigation systems based on visual information that can operate in periods or circumstances where the GPS is unavailable. Cameras are well suited for their use in embedded systems, and also can be used for estimating the target relative position.

### 1.1. Related Work

In this work, the use of a cooperative visual-based SLAM scheme is proposed, for addressing the problem of estimating the relative position of the aerial robots with respect to one target, to control the flight formation with respect to the same target.

In the literature, different approaches can be found that use visual information to carry out the control of UAVs (Visual-Servoing): [15–19]. In addition, Refs. [20–23] are examples of approaches where the position of a target is estimated in a probabilistic manner by means of the fusion of monocular measurements from two or more cameras. These works are only focused on the estimation of the target position and assume that there exists an ideal control scheme that lead the robots towards the moving target. In [24–28], different control schemes are presented for addressing the problem of maintaining a desired flight formation with respect to a moving target. A compilation of different techniques for addressing the flight formation control problem is presented in [29]. One of the underlying assumptions in these works is that the position of vehicles is available, either globally referenced or locally referenced.

In [30–32], methods for the tracking, control, and estimation of a target using a UAV with a monocular camera on board are presented. In these works, it is assumed that the geometry of the target is known or its movement is restricted to a plane. In [33,34], tracking control schemes using multi-robot and single-robot configurations are presented. These works make the strong assumption that the relative distances of the vehicles with respect to the target are known. In [35], a scheme for tracking control and estimation of a target using a small UAVs with monocular cameras on-board is presented. In this work, it is assumed that there are available measurements of the altitude of the vehicles with respect to the target. Another single-robot scheme is presented in [36]. This work relies on stereo-vision; however, as it will be shown later, the use of this sensor is unreliable in the present problem that authors state. In [37], a multi-robot system based on an LiDAR is presented.

In [38], a method to control an space robot that follows an uncooperative target using a vision system is presented. The estimation of the position of the target is carried out using an adaptive unscented Kalman filter based on measurements obtained from an homography. In this case, to obtain the homography, the coplanar visual landmarks need to be extracted from the target.

In [39], a single-robot SLAM scheme, where the state of the dynamic target is included in the system state, is presented. In this case, the position of the robot, the map, and the target are estimated using a constrained local submap filter (CLSF) based on an EKF (Extended Kalman Filter) configuration. In [40], the problem of cooperative localization and target tracking with a team of moving robots is addressed. The problem is modeled as a least-squares minimization problem and solved using sparse optimization. Static known landmarks are used to define a common reference frame for the robots and the targets. However, in many applications, it is complicated or even impossible to have prior knowledge about the position of landmarks. In [41], an approach is presented for estimating the position of objects tracked by a team of mobile robots and the use of such objects for a better self-localization. In this case, relationship among objects is used to estimate objects' position given a map. However, the geometric knowledge of objects is required in order to carry out the location of the robots. In [42], a control scheme for commanding the formation with respect to a target is presented. In this case, the local sensor information is used to estimate the relative position and orientation of the robots with respect to the target (target-centric configuration).

## 1.2. Objectives and Contributions

Recently, some works, such as [43], present the development of observers and controllers for relative estimation and circumnavigation of a target using bearing-only measurements or range with bearing measurements. In that work, the movement of the target is restricted to a plane. In addition, one of the assumptions is that range measurements with respect to the target are available, which is very difficult to obtain for non-cooperative targets. In [44], a Gaussian sum FIR filter (GSFF) to estimate the position of a target is presented. In this work, both estimation and tracking of the target are carried out only in a two-dimensional space. In [45], a UAV-based target tracking and its recognition system are presented. In this work, a geographic information system (GIS) is used to provide geo-location, environmental, and contextual information. The work in [46] presents a distributed control strategy applied to a team of agents to create a prescribed formation with its centroid tracking a moving target. In this work, one of the assumptions is that some relative position measurements between agents and the target are available.

To try avoiding the restrictions given in the previous approaches, in this present work, the use of a cooperative visual-based SLAM scheme is studied for addressing the problem of estimating the relative position of the aerial robots with respect to a non-cooperative dynamic target in GPS-denied environments. The general idea is to use a priori unknown static natural landmarks, randomly distributed in the environment, as reference points for locating UAVs with respect to a dynamic target moving freely in 3D space. The above objective is achieved using only monocular measurements of the target and landmarks, as well as measurements of altitude differential among UAVs. The proposed scheme by authors does not need any geometric knowledge of the moving target nor the landmarks. In addition, there are no assumptions about some cooperative behavior of the target.

The configuration of the proposed cooperative visual-based SLAM system is based on the standard EKF-SLAM (Extended Kalman Filter SLAM) methodology. In this context, it is extremely important to provide the proposed system with properties such as observability and robustness to ill-constrained initial conditions. The above properties have a fundamental role in the convergence of the filter, as shown in [47,48]. Therefore, an extensive nonlinear observability test is carried out in order to analyze the system. In this case, novel, contributive and important theoretical results are presented from this analysis.

In this work, two innovations are proposed to improve the observability properties of the system, and thus its performance. Firstly, the use of a target-centric SLAM configuration is proposed instead of

the use of a more common World-centric or Robot-centric SLAM configuration. In the target-centric SLAM configuration, the system state is parameterized with respect to the target position. Secondly, measurements of altitude differential between pairs of UAVs obtained from altimeters are integrated into the system.

Another key difference from most related works is presented; typically, those works focus only on addressing one side of the dual nature of the estimation-control problem. In this work, additionally to the proposed SLAM estimation technique, a flight formation control scheme based on a back-stepping approach [49] is also proposed, to allow carrying out the formation of quadcopters with respect to the target. In this case, the stability of control laws is proved using the Lyapunov theory. In simulations, the state estimated by the SLAM system is used as a feedback to the proposed control scheme to test the closed-loop performance of both estimator and control.

### 1.3. Paper Outline

The document is organized in the following manner: Section 2 presents the general system specifications and mathematical models. Section 3 presents the nonlinear observability analysis. In Section 4, the proposed SLAM method is described. Section 5 presents the proposed control system. Section 6 shows the numerical simulations results, and finally, in Section 7, the conclusions and final remarks of this work are given.

## 2. System Specification

In this section, the mathematical models used in this work are introduced. Firstly, the model used for representing the dynamics of the relative position of the *j*-th UAV-camera system with respect to the target is described. Then, the model representation of the relative position of the *i*-th landmark with respect to the target is described. In addition, the measurement models used in this work are presented: (i) the camera projection model and (ii) the relative altitude model. Finally, the dynamic model of a quadcopter is presented as well.

In applications like aerial vehicles, the attitude and heading (roll, pitch, and yaw) estimation is well handled by available systems (e.g., [50,51]). In particular, in this work, it is assumed that the orientation of the camera is always pointing toward the ground. Considering the above assumption, the system state can be simplified by removing the variables related to attitude and heading (which are provided by the AHRS). In addition, therefore, the problem can be focused on relative position estimation. In addition, it is important to note that, with this assumption, the mathematical formulation is kept simple enough to make the observability analysis of Section 3 a tractable problem. In practice, the looking-downward camera assumption can be easily addressed, for instance, with the use of a servo-controlled camera gimbal.

Regarding the visual sensors required for implementing the proposed system, the only assumption is that a set of visual features of the environment, and one visual feature of the target, can be detected and tracked consistently using any available computer vision algorithm.

### 2.1. Dynamics of the System

Let consider the following continuous-time model describing the dynamics of the proposed system (see Figure 1):

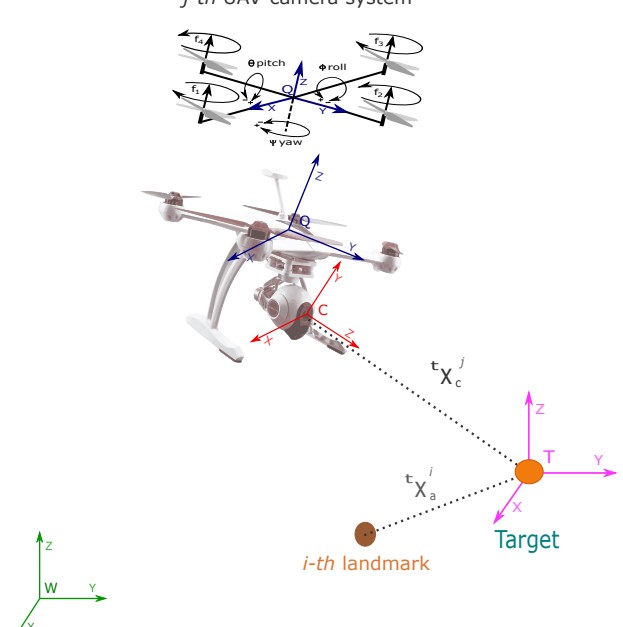

**Figure 1.** Coordinate reference systems.

$$\dot{\mathbf{x}} = \begin{bmatrix} {}^{t}\dot{\mathbf{x}}_{\mathbf{c}}{}^{1} \\ {}^{t}\dot{\mathbf{v}}_{\mathbf{c}}{}^{1} \\ \vdots \\ {}^{t}\dot{\mathbf{x}}_{\mathbf{c}}{}^{j} \\ {}^{t}\dot{\mathbf{v}}_{\mathbf{c}}{}^{j} \\ \dot{\mathbf{v}}_{\mathbf{t}} \\ {}^{t}\dot{\mathbf{x}}_{\mathbf{a}}{}^{1} \\ \vdots \\ {}^{t}\dot{\mathbf{x}}_{\mathbf{a}}{}^{i} \end{bmatrix} = \begin{bmatrix} \dot{\mathbf{x}}_{\mathbf{c}}^{1} - \dot{\mathbf{x}}_{\mathbf{t}} \\ \dot{\mathbf{v}}_{\mathbf{c}}^{1} - \dot{\mathbf{v}}_{\mathbf{t}} \\ \vdots \\ \dot{\mathbf{x}}_{\mathbf{c}}^{j} - \dot{\mathbf{x}}_{\mathbf{t}} \\ \dot{\mathbf{v}}_{\mathbf{c}}^{j} - \dot{\mathbf{v}}_{\mathbf{t}} \\ \mathbf{0}_{3\times 1} \\ \dot{\mathbf{x}}_{\mathbf{a}}^{1} - \dot{\mathbf{x}}_{\mathbf{t}} \\ \vdots \\ \dot{\mathbf{x}}_{\mathbf{a}}^{i} - \dot{\mathbf{x}}_{\mathbf{t}} \end{bmatrix} = \begin{bmatrix} \mathbf{v}_{\mathbf{c}}^{1} - \mathbf{v}_{\mathbf{t}} \\ \mathbf{0}_{3\times 1} \\ \vdots \\ \mathbf{v}_{\mathbf{c}}^{j} - \mathbf{v}_{\mathbf{t}} \\ \mathbf{0}_{3\times 1} \\ \mathbf{0}_{3\times 1} \\ \mathbf{v}_{\mathbf{a}}^{1} - \mathbf{v}_{\mathbf{t}} \\ \vdots \\ \mathbf{v}_{\mathbf{a}}^{i} - \mathbf{v}_{\mathbf{t}} \end{bmatrix} = \begin{bmatrix} {}^{t}\mathbf{v}_{\mathbf{c}}^{1} \\ \mathbf{0}_{3\times 1} \\ \vdots \\ {}^{t}\mathbf{v}_{\mathbf{c}}^{j} \\ \mathbf{0}_{3\times 1} \\ \mathbf{0}_{3\times 1} \\ -\mathbf{v}_{\mathbf{t}} \\ \vdots \\ -\mathbf{v}_{\mathbf{t}} \end{bmatrix} \tag{1}$$

where the state vector $\mathbf{x}$ is defined by:

$$\mathbf{x} = \begin{bmatrix} {}^{t}\mathbf{x}_{\mathbf{c}}^{1} & {}^{t}\mathbf{v}_{\mathbf{c}}^{1} & \cdots & {}^{t}\mathbf{x}_{\mathbf{c}}^{j} & {}^{t}\mathbf{v}_{\mathbf{c}}^{j} & \mathbf{v}_{\mathbf{t}} & {}^{t}\mathbf{x}_{\mathbf{a}}^{1} & \cdots & {}^{t}\mathbf{x}_{\mathbf{a}}^{i} \end{bmatrix}^{\mathrm{T}} \tag{2}$$

with $i = 1, ..., n_1$ and $j = 1, ..., n_2$, where $n_1$ and $n_2$ are respectively the number of landmarks included into the map and the number of UAV-camera systems.

Additionally, let $\mathbf{x_t} = \begin{bmatrix} x_t & y_t & z_t \end{bmatrix}^{\mathrm{T}}$ represent the position (in meters) of the target, with respect to the reference system $W$. Let $\mathbf{x_c}^{j} = \begin{bmatrix} x_c^{j} & y_c^{j} & z_c^{j} \end{bmatrix}^{\mathrm{T}}$ represent the position (in meters) of the reference system $C$ of the $j$-th camera, with respect to the reference system $W$. Let $\mathbf{v_t} = \begin{bmatrix} \dot{x}_t & \dot{v}_t & \dot{z}_t \end{bmatrix}^{\mathrm{T}}$ represent the linear velocity (in $\frac{m}{s}$) of the target, with respect to the reference system $W$. Let $\mathbf{v_c}^{j} = \begin{bmatrix} \dot{x}_c^{j} & \dot{y}_c^{j} & \dot{z}_c^{j} \end{bmatrix}^{\mathrm{T}}$ represent the linear velocity (in $\frac{m}{s}$) of the $j$-th camera, with respect to the reference system $W$. Let $\mathbf{x_a}^{i} = \begin{bmatrix} x_a^{i} & y_a^{i} & z_a^{i} \end{bmatrix}^{\mathrm{T}}$ be the position (in meters) of the $i$-th landmark with respect to the reference system $W$, defined by its Euclidean parameterization. Let $\mathbf{v_a}^{i} = \begin{bmatrix} \dot{x}_a^{i} & \dot{y}_a^{i} & \dot{z}_a^{i} \end{bmatrix}^{\mathrm{T}}$ represent the linear velocity (in $\frac{m}{s}$) of the $i$-th landmark, with respect to the reference system $W$. Let $\mathbf{x_c}^{j} = \begin{bmatrix} {}^{t}x_c^{j} & {}^{t}y_c^{j} & {}^{t}z_c^{j} \end{bmatrix}^{\mathrm{T}}$ represent the relative position (in meters) of the $j$-th camera with respect

to the reference system $T$. Let $\mathbf{{}^t x_a}^i = \begin{bmatrix} {}^t x_a^i & {}^t y_a^i & {}^t z_a^i \end{bmatrix}^{\mathrm{T}}$ represent the relative position (in meters) of the $i$-th landmark with respect to the reference system $T$. Finally, let $\mathbf{{}^t v_c}^j = \begin{bmatrix} {}^t \dot{x}_c^j & {}^t \dot{y}_c^j & {}^t \dot{z}_c^j \end{bmatrix}^{\mathrm{T}}$ represent the relative linear velocity (in $\frac{m}{s}$) of the $j$-th camera with respect to the reference system $T$. Figure 1 illustrates the coordinate reference systems used in this work. In Equation (1), each UAV-camera, as well as the target, is assumed to move freely in the three-dimensional space. Let note that a non-acceleration kinematic model is assumed for the UAV-camera systems and the target. In addition, note that the landmarks are assumed to remain static. Given the above $\mathbf{v_a}^i = \mathbf{0}$.

*2.2. Camera Measurement Model for Landmarks*

Let us consider the projection of a single landmark over the image plane of a camera. Using the pinhole model [52], the following expression is defined:

$$ {}^i\mathbf{z_c}^j = {}^i\mathbf{h_c}^j = \begin{bmatrix} {}^i u_c^j \\ {}^i v_c^j \end{bmatrix} = \frac{1}{{}^i z_d^j} \begin{bmatrix} \frac{f_c^j}{d_u^j} & 0 \\ 0 & \frac{f_c^j}{d_v^j} \end{bmatrix} \begin{bmatrix} {}^i x_d^j \\ {}^i y_d^j \end{bmatrix} + \begin{bmatrix} c_u^j + d_{ur}^j + d_{ut}^j \\ c_v^j + d_{vr}^j + d_{vt}^j \end{bmatrix} \tag{3} $$

Let ${}^i u_c^j$ and ${}^i v_c^j$ define the coordinates (in pixels) of the projection of the $i$-th landmark over the image of the $j$-th camera. Let $f_c^j$ be the focal length (in meters) of the $j$-th camera. Let $d_u^j$ and $d_v^j$ be the conversion parameters (in $m/pixel$) for the $j$-th camera. Let $c_u^j$ and $c_v^j$ be the coordinates (in pixels) of the image central point of the $j$-th camera. Let $d_{ur}^j$ and $d_{vr}^j$ be components (in pixels) accounting for the radial distortion of the $j$-th camera. Let $d_{ut}^j$ and $d_{vt}^j$ be components (in pixels) accounting for the tangential distortion of the $j$-th camera. All the intrinsic parameters of the $j$-th camera are assumed to be known by means of some calibration method. Let ${}^i\mathbf{p_d}^j = \begin{bmatrix} {}^i x_d^j & {}^i y_d^j & {}^i z_d^j \end{bmatrix}^{\mathrm{T}}$ represent the position (in meters) of the $i$-th landmark with respect to the coordinate reference system $C$ of the $j$-th camera. Additionally,

$$ {}^i\mathbf{p_d}^j = {}^W\mathbf{R_c}^j \left( {}^t\mathbf{x_a}^i - {}^t\mathbf{x_c}^j \right) \tag{4} $$

where ${}^W\mathbf{R_c}^j \in SO3$ is the rotation matrix that transforms from the world coordinate reference system $W$ to the coordinate reference system $C$ of the $j$-th camera. Recall that the rotation matrix ${}^W\mathbf{R_c}^j$ is known and constant assuming the use of the servo-controlled camera gimbal.

*2.3. Camera Measurement Model for the Target*

Let consider the projection of the target over the image plane of a camera. In this case, it is assumed that some visual feature points can be extracted from the target by means of some available computer vision algorithm like [53–58]. Using the pinhole model, the following expression is defined:

$$ \mathbf{{}^t z_c}^j = \mathbf{{}^t h_c}^j = \begin{bmatrix} {}^t u_c^j \\ {}^t v_c^j \end{bmatrix} = \frac{1}{{}^t z_d^j} \begin{bmatrix} \frac{f_c^j}{d_u^j} & 0 \\ 0 & \frac{f_c^j}{d_v^j} \end{bmatrix} \begin{bmatrix} {}^t x_d^j \\ {}^t y_d^j \end{bmatrix} + \begin{bmatrix} c_u^j + d_{ur}^j + d_{ut}^j \\ c_v^j + d_{vr}^j + d_{vt}^j \end{bmatrix} \tag{5} $$

Let $\mathbf{{}^t p_d}^j = \begin{bmatrix} {}^t x_d^j & {}^t y_d^j & {}^t z_d^j \end{bmatrix}^{\mathrm{T}}$ represent the position (in meters) of the target with respect to the coordinate reference system $C$ of the $j$-th camera. Additionally,

$$ \mathbf{{}^t p_d}^j = {}^W\mathbf{R_c}^j \left( -{}^t\mathbf{x_c}^j \right) \tag{6} $$

## 2.4. Altitude Differential Measurement Model

In this work, the altitude differential between a pair of UAVs will be used as a filter update measurement. In this case, the altitude differential between the *j*-th UAV and the *n*-th UAV, given by a pair of on-board altimeters in each vehicle, is defined by:

$$^n z_a^j = {}^n h_a^j = {}^t z_c^n - {}^t z_c^j \tag{7}$$

## 2.5. Dynamic Model of the Quadcopter

This section presents the dynamic model of a quadcopter which is composed of a rigid structure and four rotors. The plant has six degrees of freedom: three for the translation and three for the rotation (see Figure 1). Using the Euler–Lagrange formalism and the parametrization with respect to the Tait–Bryan angles, the model can be defined similarly as [59]:

$$\ddot{\mathbf{x}}_{\mathbf{q}}^j = \begin{bmatrix} \ddot{x}_q^j \\ \ddot{y}_q^j \\ \ddot{z}_q^j \end{bmatrix} = \frac{\mu^j}{m^j} \begin{bmatrix} \sin(\psi^j)\sin(\phi^j) + \cos(\psi^j)\sin(\theta^j)\cos(\phi^j) \\ \sin(\psi^j)\sin(\theta^j)\cos(\phi^j) - \cos(\psi^j)\sin(\phi^j) \\ \cos(\theta^j)\cos(\phi^j) - \frac{g \cdot m^j}{u_q^j \cdot \mu^j} \end{bmatrix} u_q^j \tag{8}$$

$$\ddot{\sigma}^j = \begin{bmatrix} \ddot{\phi}^j \\ \ddot{\theta}^j \\ \ddot{\psi}^j \end{bmatrix} = \mathbf{g}^j + \mathbf{B}^j \boldsymbol{\tau}^j = \begin{bmatrix} \left(\frac{I_y^j - I_z^j}{I_x^j}\right)\dot{\theta}^j\dot{\psi}^j + \left(\frac{J_p^j}{I_x^j}\right)\dot{\theta}^j\Omega^j \\ \left(\frac{I_x^j - I_z^j}{I_y^j}\right)\dot{\phi}^j\dot{\psi}^j + \left(\frac{J_p^j}{I_y^j}\right)\dot{\phi}^j\Omega^j \\ \left(\frac{I_x^j - I_y^j}{I_z^j}\right)\dot{\phi}^j\dot{\theta}^j \end{bmatrix} + \begin{bmatrix} \frac{\mu^j}{I_x^j} & 0 & 0 \\ 0 & \frac{\mu^j}{I_y^j} & 0 \\ 0 & 0 & \frac{d^j}{I_z^j} \end{bmatrix} \begin{bmatrix} \tau_\phi^j \\ \tau_\theta^j \\ \tau_\psi^j \end{bmatrix} \tag{9}$$

Let $\mathbf{x_q}^j = [x_q^j, y_q^j, z_q^j]^T$ be the position (in meters) of the *j*-th quadcopter, with respect to the reference system $W$. Let $\boldsymbol{\sigma}^j = [\phi^j, \theta^j, \psi^j]^T$ be the Tait–Bryan angles (in radians) of the *j*-th quadcopter, with respect to the reference system $W$. Let $m^j$ be the mass (in kg) of the *j*-th quadcopter. Let $\mu^j$ be the thrust factor (in N· s$^2$) of the *j*-th quadcopter. Let $g$ be the constant of gravity (in m/s$^2$). Let $u_q^j$ be the total force (in $N$) of the *j*-th quadcopter, supplied by the four rotors over the axis $z$ with respect to the coordinate reference system $Q$. Let $J_p^j$ be total inertia moment (in N· m · s$^2$) of the *j*-th quadcopter, due to the rotors. Let $d^j$ be the drag factor (in N· m · s$^2$) of the *j*-th quadcopter. Let $I_x^j$, $I_y^j$, and $I_z^j$ be the inertia moment over each axis (in N· m · s$^2$) of the *j*-th quadcopter. Let $\tau_\phi^j$, $\tau_\theta^j$, and $\tau_\psi^j$ be the torque (in $N \cdot m$) of the *j*-th quadcopter, supplied by the rotors over each axis. Finally, let $\Omega^j$ be the overall propeller rotational speed (in rad/s) of the *j*-th quadcopter.

The inputs are defined by:

$$\begin{bmatrix} u_q^j \\ \tau_\phi^j \\ \tau_\theta^j \\ \tau_\psi^j \end{bmatrix} = \begin{bmatrix} 1 & 1 & 1 & 1 \\ -l^j & 0 & l^j & 0 \\ 0 & l^j & 0 & -l^j \\ 1 & -1 & 1 & -1 \end{bmatrix} \begin{bmatrix} f_1^j \\ f_2^j \\ f_3^j \\ f_4^j \end{bmatrix} \tag{10}$$

where

$$f_m^j = \left(w_m^j\right)^2 \tag{11}$$

$$\Omega^j = w_1^j - w_2^j + w_3^j - w_4^j \tag{12}$$

In addition, $m = \{1, ..., 4\}$. Let $l^j$ be the distance (in $m$) from the center of mass of the *j*-th quadcopter to the center of one rotor. Let $f_m^j$ be the thrust force (in $N$), supplied for the $m$ rotor of the *j*-th quadcopter. In addition, let $w_m^j$ be the angular velocity (in rad/s) of the $m$ rotor of the *j*-th quadcopter.

## 3. Observability Analysis

In this section, the nonlinear observability properties of the proposed system are studied. Observability is an inherent property of a dynamic system and has an important role in the accuracy and stability of its estimation process; moreover, this fact has important consequences in the context of SLAM and the convergence of the EKF.

A system is defined as observable if the initial state $\mathbf{x_0}$, at any initial time $t_0$, can be determined given the state transition $\dot{\mathbf{x}} = \mathbf{f(x)}$, the observation model $\mathbf{y} = \mathbf{h(x)}$ of the system, and observations $\mathbf{z}[t_0, t]$ from time $t_0$ to a finite time $t$. In Hermann and Krener [60], it is demonstrated that a nonlinear system is *locally weakly observable* if the observability rank condition $rank(\mathcal{O}) = dim(\mathbf{x})$ is verified, where $\mathcal{O}$ is the observability matrix.

### 3.1. Observability Matrix

The observability matrix $\mathcal{O}$ can be computed from:

$$\mathcal{O} = \left[ \begin{array}{ccccccccc} \frac{\partial\left(\mathcal{L}_{\mathbf{f}}^0\left({}^i\mathbf{h_c}^j\right)\right)}{\partial\mathbf{x}} & \frac{\partial\left(\mathcal{L}_{\mathbf{f}}^1\left({}^i\mathbf{h_c}^j\right)\right)}{\partial\mathbf{x}} & \cdots & \frac{\partial\left(\mathcal{L}_{\mathbf{f}}^0\left({}^t\mathbf{h_c}^j\right)\right)}{\partial\mathbf{x}} & \frac{\partial\left(\mathcal{L}_{\mathbf{f}}^1\left({}^t\mathbf{h_c}^j\right)\right)}{\partial\mathbf{x}} & \cdots & \frac{\partial\left(\mathcal{L}_{\mathbf{f}}^0\left({}^n h_a^j\right)\right)}{\partial\mathbf{x}} & \frac{\partial\left(\mathcal{L}_{\mathbf{f}}^1\left({}^n h_a^j\right)\right)}{\partial\mathbf{x}} & \cdots \end{array} \right]^{\mathrm{T}} \tag{13}$$

where $\mathcal{L}_{\mathbf{f}}^s\mathbf{h}$ is the *s-th*-order Lie Derivative [61], of the scalar field $\mathbf{h}$ with respect to the vector field $\mathbf{f}$. In this work, for computing Equation (13), zero-order and first-order Lie derivatives will be used for each kind of measurement.

In case of the measurements given by a monocular camera, according to Equations (3) and (1), the following zero-order Lie derivative is defined for the projections of landmarks:

$$\frac{\partial\left(\mathcal{L}_{\mathbf{f}}^0\left({}^i\mathbf{h_c}^j\right)\right)}{\partial\mathbf{x}} = \left[ \begin{array}{ccccccccc} \mathbf{0}_{2\times6(j-1)} & -{}^i\mathbf{H_c}^j \cdot {}^W\mathbf{R_c}^j & \mathbf{0}_{2\times3} & \mathbf{0}_{2\times6(n_2-j)} & | & \mathbf{0}_{2\times3} & | & \mathbf{0}_{2\times3(i-1)} & {}^i\mathbf{H_c}^j \cdot {}^W\mathbf{R_c}^j & \mathbf{0}_{2\times3(n_1-i)} \end{array} \right] \tag{14}$$

where

$${}^i\mathbf{H_c}^j = \frac{f_c^j}{\left({}^iz_d^j\right)^2} \begin{bmatrix} \frac{{}^iz_d^j}{d_u^j} & 0 & -\frac{{}^ix_d^j}{d_u^j} \\[2mm] 0 & \frac{{}^iz_d^j}{d_v^j} & -\frac{{}^iy_d^j}{d_v^j} \end{bmatrix} \tag{15}$$

For the same kind of measurement, the following first-order Lie derivative can be defined:

$$\frac{\partial\left(\mathcal{L}_{\mathbf{f}}^1\left({}^i\mathbf{h_c}^j\right)\right)}{\partial\mathbf{x}} = \left[ \begin{array}{ccccccccc} \mathbf{0}_{2\times6(j-1)} & {}^i\mathbf{H_{dc}}^j & -{}^i\mathbf{H_c}^j \cdot {}^W\mathbf{R_c}^j & \mathbf{0}_{2\times6(n_2-j)} & | & {}^i\mathbf{H_c}^j \cdot {}^W\mathbf{R_c}^j & | & \mathbf{0}_{2\times3(i-1)} & -{}^i\mathbf{H_{dc}}^j & \mathbf{0}_{2\times3(n_1-i)} \end{array} \right] \tag{16}$$

where

$${}^i\mathbf{H_{dc}}^j = \left[ \begin{array}{ccc} {}^i\mathbf{H_1}^j & {}^i\mathbf{H_2}^j & {}^i\mathbf{H_3}^j \end{array} \right] \left({}^W\mathbf{R_c}^j\right)^2 \left({}^t\mathbf{v_c}^j + \mathbf{v_t}\right) \tag{17}$$

and

$${}^i\mathbf{H_1}^j = \frac{f_c^j}{d_u^j\left({}^iz_d^j\right)^2} \begin{bmatrix} 0 & 0 & -1 \\ 0 & 0 & 0 \end{bmatrix} \qquad {}^i\mathbf{H_2}^j = \frac{f_c^j}{d_v^j\left({}^iz_d^j\right)^2} \begin{bmatrix} 0 & 0 & 0 \\ 0 & 0 & -1 \end{bmatrix} \qquad {}^i\mathbf{H_3}^j = \frac{f_c^j}{\left({}^iz_d^j\right)^3} \begin{bmatrix} -\frac{{}^iz_d^j}{d_u^j} & 0 & \frac{2\left({}^ix_d^j\right)}{d_u^j} \\[2mm] 0 & -\frac{{}^iz_d^j}{d_v^j} & \frac{2\left({}^iy_d^j\right)}{d_v^j} \end{bmatrix} \tag{18}$$

In case of the measurement given by a monocular camera, according to Equatons (5) and (1), the following zero-order Lie derivative is defined for the projections of the target:

$$\frac{\partial\left(\mathcal{L}_{\mathbf{f}}^0\left({}^t\mathbf{h_c}^j\right)\right)}{\partial\mathbf{x}} = \left[ \begin{array}{cccccccc} \mathbf{0}_{2\times6(j-1)} & -{}^t\mathbf{H_c}^j \cdot {}^W\mathbf{R_c}^j & \mathbf{0}_{2\times3} & \mathbf{0}_{2\times6(n_2-j)} & | & \mathbf{0}_{2\times3} & | & \mathbf{0}_{2\times3n_1} \end{array} \right] \tag{19}$$

where

$$
{}^{t}\mathbf{H_c}^{j} = \frac{f_c^{j}}{\left({}^{t}z_d^{j}\right)^2}
\begin{bmatrix}
\dfrac{{}^{t}z_d^{j}}{d_u^{j}} & 0 & -\dfrac{{}^{t}x_d^{j}}{d_u^{j}} \\[2mm]
0 & \dfrac{{}^{t}z_d^{j}}{d_v^{j}} & -\dfrac{{}^{t}y_d^{j}}{d_v^{j}}
\end{bmatrix}
\tag{20}
$$

For the same kind of measurement, the following first-order Lie derivative can be defined:

$$
\frac{\partial \left( \mathcal{L}_{\mathbf{f}}^{1}\left({}^{t}\mathbf{h_c}^{j}\right)\right)}{\partial \mathbf{x}} = \begin{bmatrix} \mathbf{0}_{2\times 6(j-1)} & {}^{t}\mathbf{H_{dc}}^{j} & -{}^{t}\mathbf{H_c}^{j}\cdot {}^{W}\mathbf{R_c}^{j} & \mathbf{0}_{2\times 6(n_2-j)} & | & \mathbf{0}_{2\times 3} & | & \mathbf{0}_{2\times 3n_1} \end{bmatrix}
\tag{21}
$$

where

$$
{}^{t}\mathbf{H_{dc}}^{j} = \begin{bmatrix} {}^{t}\mathbf{H_1}^{j} & {}^{t}\mathbf{H_2}^{j} & {}^{t}\mathbf{H_3}^{j} \end{bmatrix} \left({}^{W}\mathbf{R_c}^{j}\right)^2 \left({}^{t}\mathbf{v_c}^{j}\right)
\tag{22}
$$

and

$$
{}^{t}\mathbf{H_1}^{j} = \frac{f_c^{j}}{d_u^{j}\left({}^{t}z_d^{j}\right)^2}\begin{bmatrix} 0 & 0 & -1 \\ 0 & 0 & 0 \end{bmatrix} \quad {}^{t}\mathbf{H_2}^{j} = \frac{f_c^{j}}{d_v^{j}\left({}^{t}z_d^{j}\right)^2}\begin{bmatrix} 0 & 0 & 0 \\ 0 & 0 & -1 \end{bmatrix} \quad {}^{t}\mathbf{H_3}^{j} = \frac{f_c^{j}}{\left({}^{t}z_d^{j}\right)^3}\begin{bmatrix} -\dfrac{{}^{t}z_d^{j}}{d_u^{j}} & 0 & \dfrac{2\left({}^{t}x_d^{j}\right)}{d_u^{j}} \\[2mm] 0 & -\dfrac{{}^{t}z_d^{j}}{d_v^{j}} & \dfrac{2\left({}^{t}y_d^{j}\right)}{d_v^{j}} \end{bmatrix}
\tag{23}
$$

In case of the measurement of the altitude differential, according to Equations (7) and (1), for the zero-order Lie derivative, if $j < n$ (the index of the $j$-th camera is lesser than the index of the $n$-th camera)

$$
\frac{\partial \left( \mathcal{L}_{\mathbf{f}}^{0}\left({}^{n}h_a^{j}\right)\right)}{\partial \mathbf{x}} = \begin{bmatrix} \mathbf{0}_{1\times 6(j-1)} & -\mathbf{M_x} & \mathbf{0}_{1\times 6(n-j-1)} & \mathbf{M_x} & \mathbf{0}_{1\times 6(n_2-n)} & | & \mathbf{0}_{1\times 3} & | & \mathbf{0}_{1\times 3n_1} \end{bmatrix}
\tag{24}
$$

On the other hand, if $j > n$ (the index of the $j$-th camera is higher than the index of the $n$-th camera), then

$$
\frac{\partial \left( \mathcal{L}_{\mathbf{f}}^{0}\left({}^{n}h_a^{j}\right)\right)}{\partial \mathbf{x}} = \begin{bmatrix} \mathbf{0}_{1\times 6(n-1)} & \mathbf{M_x} & \mathbf{0}_{1\times 6(j-n-1)} & -\mathbf{M_x} & \mathbf{0}_{1\times 6(n_2-j)} & | & \mathbf{0}_{1\times 3} & | & \mathbf{0}_{1\times 3n_1} \end{bmatrix}
\tag{25}
$$

and for both cases

$$
\mathbf{M_x} = \begin{bmatrix} \mathbf{0}_{1\times 2} & 1 & \mathbf{0}_{1\times 3} \end{bmatrix}
\tag{26}
$$

For the first-order Lie derivative, if $j < n$:

$$
\frac{\partial \left( \mathcal{L}_{\mathbf{f}}^{1}\left({}^{n}h_a^{j}\right)\right)}{\partial \mathbf{x}} = \begin{bmatrix} \mathbf{0}_{1\times 6(j-1)} & -\mathbf{M_{dx}} & \mathbf{0}_{1\times 6(n-j-1)} & \mathbf{M_{dx}} & \mathbf{0}_{1\times 6(n_2-n)} & | & \mathbf{0}_{1\times 3} & | & \mathbf{0}_{1\times 3n_1} \end{bmatrix}
\tag{27}
$$

In addition, if $j > n$:

$$
\frac{\partial \left( \mathcal{L}_{\mathbf{f}}^{1}\left({}^{n}h_a^{j}\right)\right)}{\partial \mathbf{x}} = \begin{bmatrix} \mathbf{0}_{1\times 6(n-1)} & \mathbf{M_{dx}} & \mathbf{0}_{1\times 6(j-n-1)} & -\mathbf{M_{dx}} & \mathbf{0}_{1\times 6(n_2-j)} & | & \mathbf{0}_{1\times 3} & | & \mathbf{0}_{1\times 3n_1} \end{bmatrix}
\tag{28}
$$

with

$$
\mathbf{M_{dx}} = \begin{bmatrix} \mathbf{0}_{1\times 5} & 1 \end{bmatrix}
\tag{29}
$$

Using the Lie derivatives described above, the observability matrix for the proposed system Equation (1) can be defined as follows:

$$
\mathcal{O} =
\begin{bmatrix}
\mathbf{0}_{2\times6(j-1)} & -{}^{i}\mathbf{H_c}^{j} \cdot {}^{W}\mathbf{R_c}^{j} & \mathbf{0}_{2\times3} & \mathbf{0}_{2\times6(n_2-j)} & | & \mathbf{0}_{2\times3} & | & \mathbf{0}_{2\times3(i-1)} & {}^{i}\mathbf{H_c}^{j} \cdot {}^{W}\mathbf{R_c}^{j} & \mathbf{0}_{2\times3(n_1-i)} \\
\mathbf{0}_{2\times6(j-1)} & {}^{i}\mathbf{H_{dc}}^{j} & -{}^{i}\mathbf{H_c}^{j} \cdot {}^{W}\mathbf{R_c}^{j} & \mathbf{0}_{2\times6(n_2-j)} & | & {}^{i}\mathbf{H_c}^{j} \cdot {}^{W}\mathbf{R_c}^{j} & | & \mathbf{0}_{2\times3(i-1)} & -{}^{i}\mathbf{H_{dc}}^{j} & \mathbf{0}_{2\times3(n_1-i)} \\
& \vdots & & & & \vdots & & & \vdots & \\
\mathbf{0}_{2\times6(j-1)} & -{}^{t}\mathbf{H_c}^{j} \cdot {}^{W}\mathbf{R_c}^{j} & \mathbf{0}_{2\times3} & \mathbf{0}_{2\times6(n_2-j)} & | & \mathbf{0}_{2\times3} & | & & \mathbf{0}_{2\times3n_1} & \\
\mathbf{0}_{2\times6(j-1)} & {}^{t}\mathbf{H_{dc}}^{j} & -{}^{t}\mathbf{H_c}^{j} \cdot {}^{W}\mathbf{R_c}^{j} & \mathbf{0}_{2\times6(n_2-j)} & | & \mathbf{0}_{2\times3} & | & & \mathbf{0}_{2\times3n_1} & \\
& \vdots & & & & \vdots & & & \vdots & \\
\mathbf{0}_{1\times6(j-1)} & -\mathbf{M_x} & \mathbf{0}_{1\times6(n-j-1)} & \mathbf{M_x} & \mathbf{0}_{1\times6(n_2-n)} & | & \mathbf{0}_{1\times3} & | & \mathbf{0}_{1\times3n_1} \\
\mathbf{0}_{1\times6(j-1)} & -\mathbf{M_{dx}} & \mathbf{0}_{1\times6(n-j-1)} & \mathbf{M_{dx}} & \mathbf{0}_{1\times6(n_2-n)} & | & \mathbf{0}_{1\times3} & | & \mathbf{0}_{1\times3n_1} \\
& \vdots & & & & \vdots & & & \vdots & \\
\mathbf{0}_{1\times6(n-1)} & \mathbf{M_x} & \mathbf{0}_{1\times6(j-n-1)} & -\mathbf{M_x} & \mathbf{0}_{1\times6(n_2-j)} & | & \mathbf{0}_{1\times3} & | & \mathbf{0}_{1\times3n_1} \\
\mathbf{0}_{1\times6(n-1)} & \mathbf{M_{dx}} & \mathbf{0}_{1\times6(j-n-1)} & -\mathbf{M_{dx}} & \mathbf{0}_{1\times6(n_2-j)} & | & \mathbf{0}_{1\times3} & | & \mathbf{0}_{1\times3n_1} \\
& \vdots & & & & \vdots & & & \vdots &
\end{bmatrix}
\tag{30}
$$

### 3.2. Theoretical Results

Three different cases of system configurations were analyzed. The idea is to study how the observability of the system is affected due to the availability (or unavailability) of the different types of measurements. The cases to be considered are:

- Case 1: The following measurements are available: (i) monocular measurements of the projection of landmarks over each UAV-camera system, (ii) multiple (two or more) monocular measurements of the projection of the target over each UAV-camera system.
- Case 2: The following measurements are available: (i) monocular measurements of projection of landmarks over each UAV-camera system, (ii) a single monocular measurement of the projection of the target over a UAV-camera system, and (iii) the altitude differential measurements between UAV-camera systems.
- Case 3: The following measurements are available: (i) monocular measurements of the projection of landmarks over each UAV-camera system, (ii) multiple (two or more) monocular measurements of the projection of the target over each UAV-camera system, and (iii) the altitude differential measurements between UAV-camera systems.

### 3.2.1. Case 1

For the first case, considering only the respective derivatives on the observability matrix, Equation (30), the maximum rank of the observability matrix $\mathcal{O}$ is $rank(\mathcal{O}) = (3n_1 + 6n_2 + 3) - 1$, where $n_1$ is the number of landmarks being measured, $n_2$ is the number of robots, and 3 is the number of states of the linear velocity of the target. In this case, $n_1$ is multiplied by 3, since this is the number of states per landmark, and $n_2$ is multiplied by 6, since this is the number of states per robot. Therefore,

$\mathcal{O}$ will be rank deficient ($rank(\mathcal{O}) < dim(\mathbf{x})$). The unobservable modes are spanned by the right nullspace basis of the observability matrix $\mathcal{O}$:

$$\mathbf{N_1} = null(\mathcal{O}) = \begin{bmatrix} {}^t\mathbf{x_c}^1 & {}^t\mathbf{v_c}^1 & \cdots & {}^t\mathbf{x_c}^j & {}^t\mathbf{v_c}^j & | & \mathbf{v_t} & | & {}^t\mathbf{x_a}^1 & \cdots & {}^t\mathbf{x_a}^i \end{bmatrix}^T = \mathbf{x} \tag{31}$$

It is straightforward to verify that the right nullspace basis of $\mathcal{O}$ spans for $\mathbf{N_1}$, (i.e., $\mathcal{O}\mathbf{N_1} = \mathbf{0}$). From Equation (31), it can be seen that the unobservable modes cross through all states; therefore, all states are unobservable. It should be noted that, adding Lie derivatives of higher-order to the observability matrix, the previous result does not improve.

### 3.2.2. Case 2

For the second case, the maximum rank of the observability matrix $\mathcal{O}$ is $rank(\mathcal{O}) = (3n_1 + 6n_2 + 3) - 2$. Therefore, $\mathcal{O}$ will be rank deficient ($rank(\mathcal{O}) < dim(\mathbf{x})$). In this case, the unobservable modes are spanned by the following right nullspace basis of the observability matrix $\mathcal{O}$:

$$\mathbf{N_2} = null(\mathcal{O}) = \begin{bmatrix} \mathbf{0}_{3\times1} & {}^t\mathbf{x_c}^j & \cdots & \mathbf{0}_{3\times1} & {}^t\mathbf{x_c}^j & | & {}^t\mathbf{x_c}^j & | & \mathbf{0}_{3\times1} & \cdots & \mathbf{0}_{3\times1} \\ {}^t\mathbf{x_c}^j & \left[{}^t\mathbf{x_c}^j \times {}^t\mathbf{v_c}^j\right]_\times \mathbf{z} & \cdots & {}^t\mathbf{x_c}^j & \left[{}^t\mathbf{x_c}^j \times {}^t\mathbf{v_c}^j\right]_\times \mathbf{z} & | & \left[{}^t\mathbf{x_c}^j \times {}^t\mathbf{v_c}^j\right]_\times \mathbf{z} & | & {}^t\mathbf{x_c}^j & \cdots & {}^t\mathbf{x_c}^j \end{bmatrix}^T \tag{32}$$

where $\mathbf{z} = \begin{bmatrix} 0 & 0 & \frac{1}{{}^{t}z_c^j} \end{bmatrix}^T$. The states ${}^t\mathbf{x_c}^j$, ${}^t\mathbf{v_c}^j$ involved in the right nullspace basis ($\mathbf{N_2}$) belong to the $j$-th robot that observes the target. It is straightforward to verify that the right nullspace basis of $\mathcal{O}$ spans for $\mathbf{N_2}$, (i.e., $\mathcal{O}\mathbf{N_2} = \mathbf{0}$). From Equation (32) it can be seen that the unobservable modes cross through all states, therefore, all states are unobservable. It should be noted that adding Lie derivatives of higher-order to the observability matrix the previous result does not improve.

### 3.2.3. Case 3

For the third case, the maximum rank of the observability matrix $\mathcal{O}$ is $rank(\mathcal{O}) = (3n_1 + 6n_2 + 3)$. Given that $dim(\mathbf{x}) = 3n_1 + 6n_2 + 3$, then, the system under the third case is *locally weakly observable* because $rank(\mathcal{O}) = dim(\mathbf{x})$.

### 3.2.4. Remarks to the Theoretical Results

The results of the observability analysis are summarized in Table 1.

**Table 1.** Results of the nonlinear observability analysis of the proposed system.

| | Monocular Measurements of Landmarks | Monocular Measurements of the Target (Single) | Monocular Measurements of the Target (Multiple) | Altitude Differential Measurements | Unobservable Modes | Unobservable States | Observable States |
|---|---|---|---|---|---|---|---|
| Case 1 | Yes | No | Yes | No | 1 | x | - |
| Case 2 | Yes | Yes | No | Yes | 2 | x | - |
| Case 3 | Yes | No | Yes | Yes | 0 | - | x |

- Having altitude differential measurements between two UAV-camera systems is a necessary condition to obtain the previous results (see Figure 2). In this case, results do not improve when adding more altitude differentials.
- It is necessary to have at least two monocular measurements of the target in order to obtain the previous results (see Figure 2).
- To obtain the previous results, it is necessary to link the members of the multi-UAV system through at least two measurements in common (see Figure 2). That means that: (i) a robot needs to share the observation of at least two landmarks with any other robot; or (ii) a robot needs to share the observation of at least one landmark and the target with any other robot; or (iii) a robot needs to share the observation of at least one landmark with any other robot having the

measurement of its altitude differential with that same robot; or (iv) a robot needs to share the observation of the target with any other robot having the measurement of its altitude differential with that same robot.

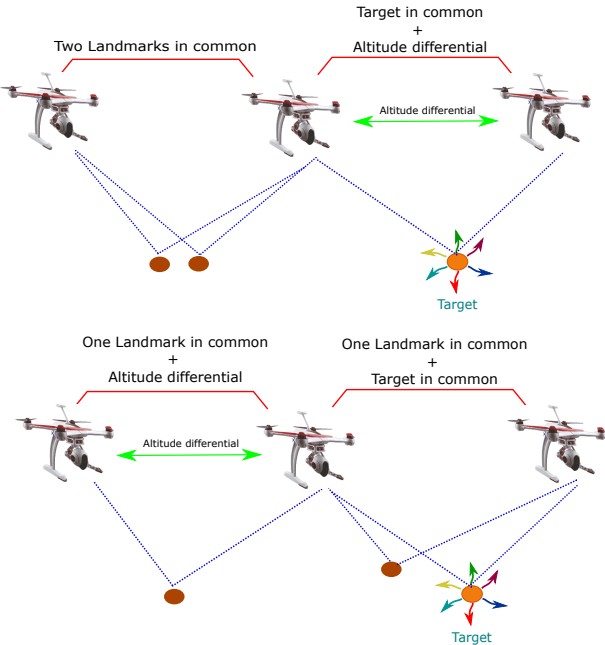

**Figure 2.** Minimum requirements to obtain the results of the observability analysis for the proposed system in Case 3.

## 4. EKF-Based SLAM

In this work, the system state in Equation (2) is estimated using the EKF-based SLAM methodology [62,63]. Figure 3 shows the architecture of the proposed system. In this case, it is assumed a ground-based centralized architecture, where all the most computing-demanding processes, like the visual processing and the main estimation process, are carried out on a ground-based computer. Algorithm 1 outlines the proposed EKF-SLAM based method.

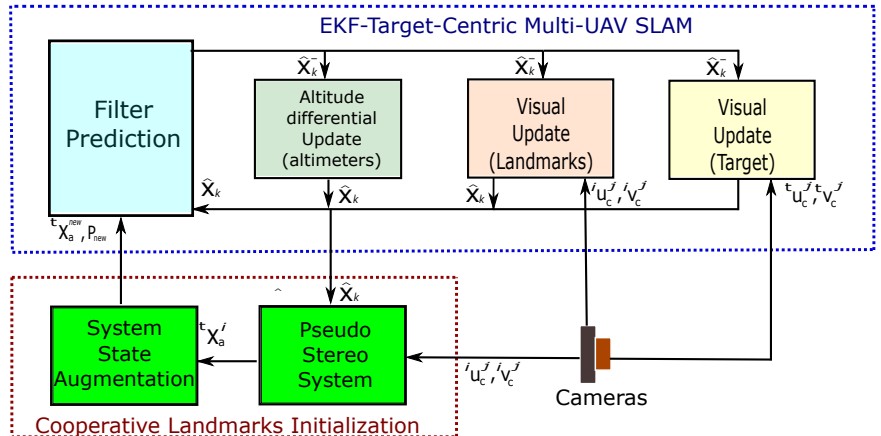

**Figure 3.** Block diagram showing the EKF-SLAM architecture of the proposed system.

---

**Algorithm 1:** EKF-SLAM target-Centric Multi-UAV

---

**Data:** $\hat{\mathbf{x}}_0, \mathbf{P}_0, \mathbf{Q}, \mathbf{R}, {}^i\mathbf{R}^j, th_1, th_2$
(where $th_1$ is an arbitrary threshold for the least number of features allowed within the map. In addition, $th_2$ is an arbitrary threshold for the greatest number of sample steps allowed without measurements for a map feature)

**Input:** $\hat{\mathbf{x}}_{k-1}, \mathbf{P}_{k-1}$

**Output:** $\hat{\mathbf{x}}_k, \mathbf{P}_k$

1 **while** *State estimation is carried out* **do**

    A. System Prediction

2     $[\hat{\mathbf{x}}_k^-, \mathbf{P}_k^-] = EKF\_Prediction(\hat{\mathbf{x}}_{k-1}, \mathbf{P}_{k-1})$

    B. Measurements Updates

    B.1.  Visual Updates (Landmarks)

3     **if** *Visual measurements of landmarks are available* **then**

4         **for** 1 *to i* **do**

5             **if** *i-th landmark was measured* **then**

6                 **for** 1 *to j* **do**

7                     **if** *i-th landmark was measured by the j-th UAV* **then**

8                         $[\hat{\mathbf{x}}_k, \mathbf{P}_k] = EKF\_Update\_Visual\_Landmarks(\hat{\mathbf{x}}_k^-, \mathbf{P}_k^-, {}^i\mathbf{h_c}^j, {}^i\mathbf{z_c}^j)$

9                     **end**

10                 **end**

11             **end**

12         **end**

13     **end**

    B.2.  Visual Updates (target)

14     **if** *Visual measurements of the target are available* **then**

15         **for** 1 *to j* **do**

16             **if** *Target was measured by the j-th UAV* **then**

17                 $[\hat{\mathbf{x}}_k, \mathbf{P}_k] = EKF\_Update\_Visual\_Target(\hat{\mathbf{x}}_k^-, \mathbf{P}_k^-, {}^t\mathbf{h_c}^j, {}^t\mathbf{z_c}^j)$

18             **end**

19         **end**

20     **end**

    B.3.  Altitude Differential Updates

21     **if** *Altitude Differential measurements between pairs of UAVs are available* **then**

22         **for** 1 *to* $n_3$ **do**

            (where $n_3$ is the number of Altitude Differential measurements)

23             $[\hat{\mathbf{x}}_k, \mathbf{P}_k] = EKF\_Update\_Altitude\_Differential(\hat{\mathbf{x}}_k^-, \mathbf{P}_k^-, {}^nh_a^j, {}^nz_a^j)$

24         **end**

25     **end**

    C. Map features initialization

26     **if** $n_1 < th_1$ **then**

27         **for** 1 *to* $n_4$ **do**

            (where $n_4$ is the number of new map features to initialize)

28             $[\mathbf{x}, \mathbf{P}_{new}] = Initialization\_New\_Map\_Features(\hat{\mathbf{x}}_k, \mathbf{P}_k, {}^iu_c^j, {}^iu_c^j, {}^i\mathbf{R}^j)$

29         **end**

30     **end**

    D. Map management

31     **for** 1 *to i* **do**

32         **if** ${}^tk_a^i > th_2$ **then**

            (where ${}^tk_a^i$ is the number of sample steps without measurements for the $i$-th landmark.

33             ${}^t\mathbf{x_a}^i = [\ ]$

34             $\mathbf{P}({}^t\mathbf{x_a}^i) = [\ ]$

35         **end**

36     **end**

37 **end**

### 4.1. System Prediction

At every step $k$, the estimated system state $\hat{\mathbf{x}}$ takes a step forward by the following discrete model:

$$
\mathbf{x}_k = \mathbf{f}(\mathbf{x}_{k-1}, \mathbf{n}_{k-1}) =
\begin{bmatrix}
{}^t\mathbf{x}_{ck}^{1} \\
{}^t\mathbf{v}_{ck}^{1} \\
\vdots \\
{}^t\mathbf{x}_{ck}^{j} \\
{}^t\mathbf{v}_{ck}^{j} \\
\mathbf{v}_{tk} \\
{}^t\mathbf{x}_{ak}^{1} \\
\vdots \\
{}^t\mathbf{x}_{ak}^{i}
\end{bmatrix}
=
\begin{bmatrix}
{}^t\mathbf{x}_{ck-1}^{1} + ({}^t\mathbf{v}_{ck-1}^{1})\Delta t \\
{}^t\mathbf{v}_{ck-1}^{1} + {}^t\boldsymbol{\eta}_{ck-1}^{1} \\
\vdots \\
{}^t\mathbf{x}_{ck-1}^{j} + ({}^t\mathbf{v}_{ck-1}^{j})\Delta t \\
{}^t\mathbf{v}_{ck-1}^{j} + {}^t\boldsymbol{\eta}_{ck-1}^{j} \\
\mathbf{v}_{tk-1} + \boldsymbol{\zeta}_{tk-1} \\
{}^t\mathbf{x}_{ak-1}^{1} - (\mathbf{v}_{tk-1})\Delta t \\
\vdots \\
{}^t\mathbf{x}_{ak-1}^{i} - (\mathbf{v}_{tk-1})\Delta t
\end{bmatrix}
\tag{33}
$$

$$
\mathbf{n}_k =
\begin{bmatrix}
{}^t\boldsymbol{\eta}_{ck}^{j} \\
\boldsymbol{\zeta}_{tk}
\end{bmatrix}
=
\begin{bmatrix}
{}^t\mathbf{a}_c^{j}\Delta t \\
\mathbf{a}_t \Delta t
\end{bmatrix}
\tag{34}
$$

where ${}^t\mathbf{a}_c^{j}$ and $\mathbf{a_t}$ represent unknown linear accelerations that are assumed to have Gaussian distribution with zero mean. Moreover, let $\mathbf{n} \sim \mathcal{N}(\mathbf{0}, \mathbf{Q})$ be the noise vector that affects the state, while $\Delta t$ is the differential of time and $k$ the sample step. In this work, a Gaussian random process is used for propagating the velocity of the vehicle. The proposed scheme is independent of the kind of aircraft and therefore is not restricted by the use of an specific dynamic model.

An Extended Kalman Filter (EKF) propagates the system state $\hat{\mathbf{x}}$ over time as follows:

$$
\hat{\mathbf{x}}_k^{-} = \mathbf{f}(\hat{\mathbf{x}}_{k-1}, \mathbf{0})
\tag{35}
$$

The state covariance matrix $P$ takes a step forward using [47]:

$$
\mathbf{P}_k^{-} = \mathbf{A}_k \mathbf{P}_{k-1} \mathbf{A}_k^{T} + \mathbf{W}_k \mathbf{Q}_{k-1} \mathbf{W}_k^{T}
\tag{36}
$$

with

$$
\mathbf{A}_k = \frac{\partial \mathbf{f}}{\partial \mathbf{x}}(\hat{\mathbf{x}}_{k-1}, \mathbf{0}) \qquad \mathbf{W}_k = \frac{\partial \mathbf{f}}{\partial \mathbf{n}}(\hat{\mathbf{x}}_{k-1}, \mathbf{0})
\tag{37}
$$

### 4.2. Measurement Updates

Assuming that, for the current sample step, a set of measurements is available, then the filter is updated with the Kalman update equations as follows [47]:

$$
\hat{\mathbf{x}}_k = \hat{\mathbf{x}}_k^{-} + \mathbf{K}_k(\mathbf{z}_k - \mathbf{h}(\hat{\mathbf{x}}_k^{-}, \mathbf{0}))
\tag{38}
$$

$$
\mathbf{P}_k = (\mathbf{I} - \mathbf{K}_k \mathbf{C}_k)\mathbf{P}_k^{-}
\tag{39}
$$

with

$$
\mathbf{K}_k = \mathbf{P}_k^{-} \mathbf{C}_k^{T}(\mathbf{C}_k \mathbf{P}_k^{-} \mathbf{C}_k^{T} + \mathbf{V}_k \mathbf{R}_k \mathbf{V}_k^{T})^{-1}
\tag{40}
$$

and

$$
\mathbf{C}_k = \frac{\partial \mathbf{h}}{\partial \mathbf{x}}(\hat{\mathbf{x}}_k^{-}, \mathbf{0}) \qquad \mathbf{V}_k = \frac{\partial \mathbf{h}}{\partial \mathbf{r}}(\hat{\mathbf{x}}_k^{-}, \mathbf{0})
\tag{41}
$$

where $\mathbf{z}$ is the vector of the current measurements, $\mathbf{h}$ is the vector of the current prediction measurements, and $\mathbf{r} \sim \mathcal{N}(\mathbf{0}, \mathbf{R})$ is the noise vector that affects the measurements.

4.2.1. Visual Updates (Landmarks)

When in the current sample step a set of visual measurements of the landmarks is available, the system is updated with this kind of measurements. In this case:

$$\mathbf{h} = \left[\begin{array}{ccc} {}^1\mathbf{h_c}^j & \cdots & {}^i\mathbf{h_c}^j \end{array}\right]^{\mathrm{T}} \qquad \mathbf{z} = \left[\begin{array}{ccc} {}^1\mathbf{z_c}^j & \cdots & {}^i\mathbf{z_c}^j \end{array}\right]^{\mathrm{T}} \tag{42}$$

4.2.2. Visual Updates (Target)

When in the current sample step a set of visual measurements of the target is available, the system is updated with this kind of measurements. In this case:

$$\mathbf{h} = \left[\begin{array}{ccc} {}^\mathsf{t}\mathbf{h_c}^1 & \cdots & {}^\mathsf{t}\mathbf{h_c}^j \end{array}\right]^{\mathrm{T}} \qquad \mathbf{z} = \left[\begin{array}{ccc} {}^\mathsf{t}\mathbf{z_c}^1 & \cdots & {}^\mathsf{t}\mathbf{z_c}^j \end{array}\right]^{\mathrm{T}} \tag{43}$$

4.2.3. Altitude Differential Updates

When in the current sample step a set of altitude differential measurements between pairs of UAVs is available, the system is updated with this kind of measurements. In this case:

$$\mathbf{h} = \left[\begin{array}{cc} {}^n h_a^j & \cdots \end{array}\right]^{\mathrm{T}} \qquad \mathbf{z} = \left[\begin{array}{cc} {}^n z_a^j & \cdots \end{array}\right]^{\mathrm{T}} \tag{44}$$

*4.3. Map Features Initialization*

The system state $\mathbf{x}$ is augmented with new map features using a cooperative strategy. The 3D relative position with respect to the target of the new map features is estimated using a pseudo-stereo system formed by the monocular cameras mounted on a pair of UAVs that observe common landmarks. In this case, when a new potential landmark is observed by two cameras, then it is initialized employing a linear triangulation.

The state of the new feature is computed using the a posteriori values obtained in the correction stage of the EKF. According to Equation (3), the following expression can be defined in homogeneous coordinates [52]:

$$^i\gamma_c^j \left[\begin{array}{c} {}^i u_c^j \\ {}^i v_c^j \\ 1 \end{array}\right] = \left[\begin{array}{cc} \mathbf{T_c}^j & \mathbf{0}_{3\times1} \end{array}\right] \hat{\mathbf{E}}_\mathbf{c}^j \left[\begin{array}{c} {}^\mathsf{t}\mathbf{x_a}^i \\ 1 \end{array}\right] \tag{45}$$

where $^i\gamma_c^j$ is a scale factor. Additionally, it is defined:

$$\hat{\mathbf{E}}_\mathbf{c}^j = \left[\begin{array}{cc} {}^W\mathbf{R_c}^j & {}^\mathsf{t}\hat{\mathbf{x}}_\mathbf{c}^j \\ \mathbf{0}_{1\times3} & 1 \end{array}\right]$$

$$\mathbf{T_c}^j = \left[\begin{array}{ccc} \frac{f_c^j}{d_u^j} & 0 & c_u^j + d_{ur}^j + d_{ut}^j \\ 0 & \frac{f_c^j}{d_v^j} & c_v^j + d_{vr}^j + d_{vt}^j \\ 0 & 0 & 1 \end{array}\right] \tag{46}$$

Using Equation (45), and considering the projection onto two any UAV-cameras, a linear system can be formed in order to estimate $^\mathsf{t}\mathbf{x_a}^i$:

$$\mathbf{D}^i \cdot {}^\mathsf{t}\mathbf{x_a}^i = \mathbf{b}^i \qquad {}^\mathsf{t}\mathbf{x_a}^i = \mathbf{D}^{i\dagger} \cdot \mathbf{b}^i \tag{47}$$

where $\mathbf{D}^{i^\dagger}$ is the Moore Penrose right pseudo-inverse matrix of $\mathbf{D}^i$, and

$$\mathbf{D}^i = \begin{bmatrix} k^j_{31} \, {}^i u^j_c - k^j_{11} & k^j_{32} \, {}^i u^j_c - k^j_{12} & k^j_{33} \, {}^i u^j_c - k^j_{13} \\ k^j_{31} \, {}^i v^j_c - k^j_{21} & k^j_{32} \, {}^i v^j_c - k^j_{22} & k^j_{33} \, {}^i v^j_c - k^j_{23} \end{bmatrix}$$

$$\mathbf{b}^i = \begin{bmatrix} k^j_{14} - k^j_{34} \, {}^i u^j_c \\ k^j_{24} - k^j_{34} \, {}^i v^j_c \end{bmatrix} \tag{48}$$

with

$$\begin{bmatrix} \mathbf{T_c}^j & \mathbf{0}_{3\times 1} \end{bmatrix} \mathbf{\hat{E}}^j_c = \begin{bmatrix} k^j_{11} & k^j_{12} & k^j_{13} & k^j_{14} \\ k^j_{21} & k^j_{22} & k^j_{23} & k^j_{24} \\ k^j_{31} & k^j_{32} & k^j_{33} & k^j_{34} \end{bmatrix} \tag{49}$$

When a new landmark is initialized, the system state $\mathbf{x}$ is augmented by:

$$\mathbf{x} = \begin{bmatrix} {}^t\mathbf{x_c}^1 & {}^t\mathbf{v_c}^1 & \cdots & {}^t\mathbf{x_c}^j & {}^t\mathbf{v_c}^j & \mathbf{v_t} & {}^t\mathbf{x_a}^1 & \cdots & {}^t\mathbf{x_a}^i & {}^t\mathbf{x_a}^{new} \end{bmatrix}^\mathrm{T} \tag{50}$$

In addition, the new covariance matrix $\mathbf{P}_{new}$ is computed by:

$$\mathbf{P}_{new} = \Delta\mathbf{J} \begin{bmatrix} \mathbf{P} & \mathbf{0} \\ \mathbf{0} & {}^i\mathbf{R}^j \end{bmatrix} = \Delta\mathbf{J}^\mathrm{T} \tag{51}$$

where $\Delta\mathbf{J}$ is the Jacobian for the initialization function, and ${}^i\mathbf{R}^j$ is the measurement noise covariance matrix for $({}^i u^j_c, {}^i v^j_c)$.

### 4.4. Map Management

It is well known that due to the nature of the Kalman Filter, in SLAM, the system state can always reach a size that will make it impossible to maintain a real-time performance for a given hardware configuration. In this sense, the present work is mainly intended to address the problem of navigation with respect to a target (local navigation). Therefore, features that are left behind by the movement of the cameras will be removed from the system state and covariance matrix. This strategy will prevent that the system state size reaches an unmanageable amount that seriously affects the computational performance.

## 5. Control System

This section presents the control scheme that allows for carrying out the flight formation of the UAVs with respect to the target. While the proposed control scheme is presented assuming that UAVs are quadcopters, the methodology can be easily extended for being used with any other aerial configurations.

### 5.1. Dynamic Model Flight Formation

The dynamic model used for representing the flight formation is based on the leader–follower scheme [64]. In this case, it is desired to maintain the $j$-th UAV to a distance ${}^t x^j_q$, ${}^t y^j_q$ (in the $x - y$ plane) from the target (See Figure 4). In addition, it is also desired to maintain the $j$-th UAV at an altitude differential ${}^t z^j_q$ from the target (See Figure 4). Given the above considerations, the following expression can be defined:

$$\mathbf{{}^t x_q}^j = \begin{bmatrix} {}^t x^j_q \\ {}^t y^j_q \\ {}^t z^j_q \end{bmatrix} = \mathbf{x_q}^j - \mathbf{x_t} \tag{52}$$

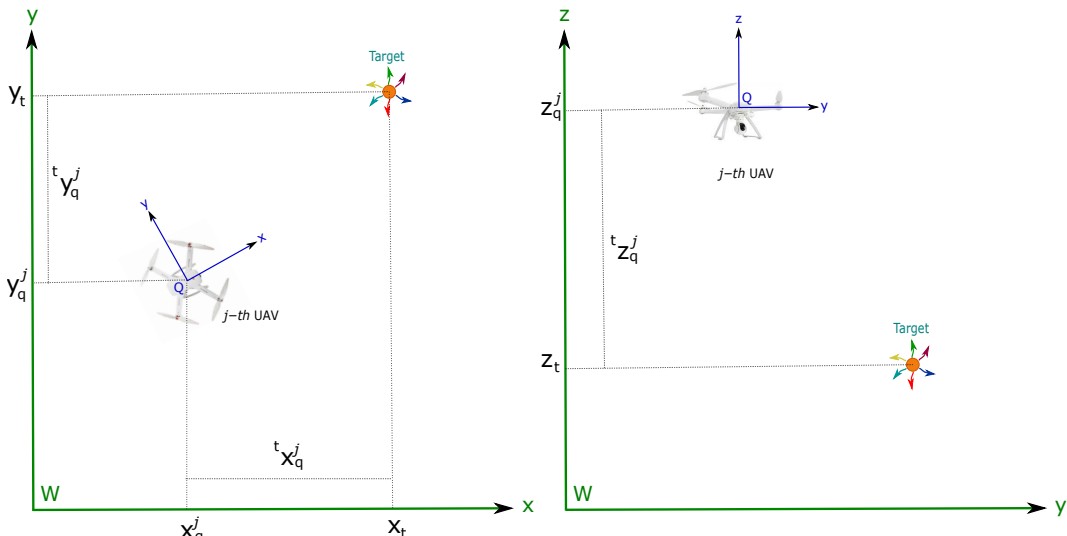

**Figure 4.** UAVs—target flight formation.

Differentiating twice Equation (52) with respect to time and using Equation (8), the dynamics of the formation ${}^{t}\mathbf{x_q}^{j}$ can be obtained:

$$
{}^{t}\ddot{\mathbf{x}}_{\mathbf{q}}{}^{j} = \ddot{\mathbf{x}}_{\mathbf{q}}^{j} - \ddot{\mathbf{x}}_{\mathbf{t}} = \frac{\mu^{j}}{m^{j}} \begin{bmatrix} \sin(\psi^{j})\sin(\phi^{j}) + \cos(\psi^{j})\sin(\theta^{j})\cos(\phi^{j}) \\ \sin(\psi^{j})\sin(\theta^{j})\cos(\phi^{j}) - \cos(\psi^{j})\sin(\phi^{j}) \\ \cos(\theta^{j})\cos(\phi^{j}) - \frac{g \cdot m^{j}}{u_{q}^{j} \cdot \mu^{j}} \end{bmatrix} u_{q}^{j} - \ddot{\mathbf{x}}_{\mathbf{t}} \tag{53}
$$

*5.2. Control Scheme*

The control scheme is designed to allow that the flight formation converges to the desired values. The control system is divided into two subsystems (see Figure 5): (i) the flight formation control, which involves the translational dynamics of the quadcopter and the target; (ii) the rotational control. Because the reference signal of the rotational subsystem is obtained from the flight formation control subsystem, it is assumed that the dynamics of the former is faster than the dynamics of the latter.

A master–slave system is defined by:

$$
\sum = \begin{cases} {}^{t}\ddot{\mathbf{x}}_{\mathbf{q}}{}^{j} = \ddot{\mathbf{x}}_{\mathbf{q}}^{j} - \ddot{\mathbf{x}}_{\mathbf{t}} \\ \ddot{\sigma}^{j} = \mathbf{g}^{j} + \mathbf{B}^{j}\boldsymbol{\tau}^{j} \end{cases} \tag{54}
$$

The overall control scheme is based on the Backstepping control technique [49], and each control-loop subsystem is designed by the so-called technique exact linearization by state feedback [49]. The state values required by the flight formation control subsystem are obtained from the estimator described in Section 4. The attitude of the *j*-th UAV is obtained assuming the availability of an on-board AHRS. To obtain the control laws using analysis in continuous time, it is assumed that the estimated values are passed through a zero-order hold (ZOH) (see Figure 5).

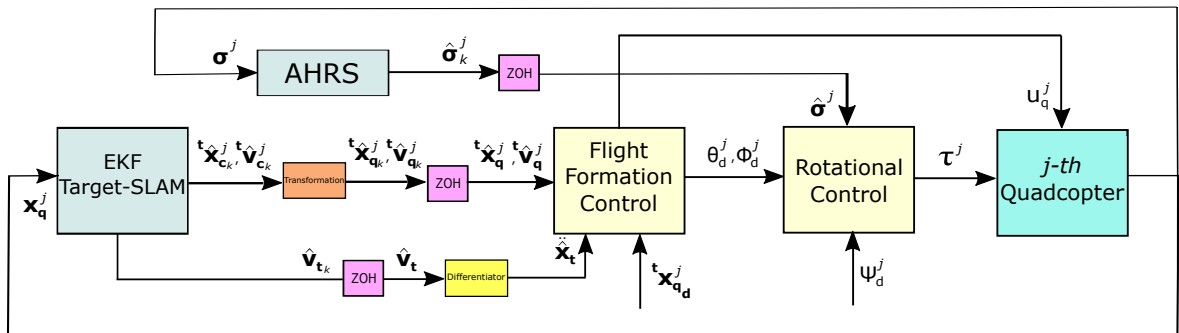

**Figure 5.** Control scheme diagram.

5.2.1. Flight Formation Subsystem Control

Since the estimated state of the relative position of the *j*-th UAV with respect to the target is defined with respect to the reference system C (see Sections 2 and 4), it is necessary to apply a transformation to the estimated state ${}^{t}\hat{\mathbf{x}}_{\mathbf{c}}{}^{j}$ for obtaining ${}^{t}\hat{\mathbf{x}}_{\mathbf{q}}{}^{j}$, which, in turn, is necessary to obtain the control laws. Therefore, the following equation is defined:

$$
{}^{t}\hat{\mathbf{x}}_{\mathbf{q}}{}^{j} = {}^{t}\hat{\mathbf{x}}_{\mathbf{c}}{}^{j} - {}^{q}\mathbf{d}_{\mathbf{c}}{}^{j} \tag{55}
$$

Let ${}^{q}\mathbf{d}_{\mathbf{c}}{}^{j}$ be the translation vector (in meters) from the coordinate reference system *Q* to the coordinate reference system *C*. Note that ${}^{q}\mathbf{d}_{\mathbf{c}}{}^{j}$ is assumed to be known and constant.

Firstly, a control law for the altitude differential ${}^{t}z_{q}^{j}$ of the flight formation subsystem is developed. Considering the dynamics of ${}^{t}z_{q}^{j}$ given in Equation (53), with control input $u_{q}^{j}$, and defining:

$$
s_{z}^{j} = \dot{e}_{z}^{j} + \lambda_{z}^{j} e_{z}^{j} + \kappa_{z}^{j} \int_{0}^{t} e_{z}^{j} dt \tag{56}
$$

where the error signal is $e_{z}^{j} = {}^{t}\hat{z}_{q}^{j} - {}^{t}z_{q_{d}}^{j}$, and ${}^{t}z_{q_{d}}^{j}$ is the desired value.

Deriving Equation (56) and substituting in Equation (53), it is obtained:

$$
\dot{s}_{z}^{j} = \left(\frac{\mu^{j}}{m^{j}}\right)\left(\cos(\hat{\theta}^{j})\cos(\hat{\phi}^{j})\right) u_{q}^{j} - g + \lambda_{z}^{j}\dot{e}_{z}^{j} + \kappa_{z}^{j} e_{z}^{j} - \ddot{z}_{t} - {}^{t}\ddot{z}_{q_{d}}^{j} \tag{57}
$$

The following control law is proposed:

$$
u_{q}^{j} = \frac{m^{j}\left({}^{t}\ddot{z}_{q_{d}}^{j} + \ddot{z}_{t} - \lambda_{z}^{j}\dot{e}_{z}^{j} - \kappa_{z}^{j} e_{z}^{j} - k_{z}^{j} s_{z}^{j} + g\right)}{\mu^{j}\left(\cos(\hat{\theta}^{j})\cos(\hat{\phi}^{j})\right)} \tag{58}
$$

where $k_{z}^{j}$, $\lambda_{z}^{j}$, and $\kappa_{z}^{j}$ are positive gains assuming that $\hat{\theta}^{j}$, $\hat{\phi}^{j} \in [-\frac{\pi}{2}, \frac{\pi}{2}]$.

Now, consider the longitudinal and lateral differential ${}^{t}x_{q}^{j}$ and ${}^{t}y_{q}^{j}$ of the flight formation subsystem. By substituting Equation (58) in Equation (53), then, for the two first states, it can be obtained:

$$
\begin{bmatrix} {}^{t}\ddot{x}_{q}^{j} \\ {}^{t}\ddot{y}_{q}^{j} \end{bmatrix} = u_{z}^{j} \begin{bmatrix} \sin(\hat{\psi}^{j}) & \cos(\hat{\psi}^{j}) \\ -\cos(\hat{\psi}^{j}) & \sin(\hat{\psi}^{j}) \end{bmatrix} \begin{bmatrix} \frac{\tan(\hat{\phi}^{j})}{\cos(\hat{\theta}^{j})} \\ \tan(\hat{\theta}^{j}) \end{bmatrix} - \begin{bmatrix} \ddot{x}_{t} \\ \ddot{y}_{t} \end{bmatrix} \tag{59}
$$

Taking $\phi^{j}$ and $\theta^{j}$ as control inputs in Equation (59), and defining:

$$
\begin{aligned}
s_{x}^{j} &= \dot{e}_{x}^{j} + \lambda_{x}^{j} e_{x}^{j} + \kappa_{x}^{j} \int_{0}^{t} e_{x}^{j} dt \\
s_{y}^{j} &= \dot{e}_{y}^{j} + \lambda_{y}^{j} e_{y}^{j} + \kappa_{y}^{j} \int_{0}^{t} e_{y}^{j} dt
\end{aligned} \tag{60}
$$

with error signals $e_x^j = {}^t\hat{x}_q^j - {}^t x_{q_d}^j$ and $e_y^j = {}^t\hat{y}_q^j - {}^t y_{q_d}^j$, being ${}^t x_{q_d}^j$ and ${}^t y_{q_d}^j$ the desired values, it is possible to derive Equation (60) and substitute in Equation (59) in order to obtain:

$$
\begin{bmatrix} \dot{s}_x^j \\ \dot{s}_y^j \end{bmatrix} = u_z^j \begin{bmatrix} \sin(\hat{\psi}^j) & \cos(\hat{\psi}^j) \\ -\cos(\hat{\psi}^j) & \sin(\hat{\psi}^j) \end{bmatrix} \begin{bmatrix} \frac{\tan(\phi^j)}{\cos(\hat{\theta}^j)} \\ \tan(\theta^j) \end{bmatrix} + \begin{bmatrix} \lambda_x^j \dot{e}_x^j + \kappa_x^j e_x^j - {}^t\ddot{x}_{q_d}^j - \ddot{x}_t \\ \lambda_y^j \dot{e}_y^j + \kappa_y^j e_y^j - {}^t\ddot{y}_{q_d}^j - \ddot{y}_t \end{bmatrix} \tag{61}
$$

The following control laws are proposed:

$$
\begin{bmatrix} \theta^j \\ \phi^j \end{bmatrix} = \arctan\left( \frac{1}{u_z^j} \begin{bmatrix} \cos(\hat{\psi}^j) & \sin(\hat{\psi}^j) \\ \sin(\hat{\psi}^j)\cos(\hat{\theta}^j) & -\cos(\hat{\psi}^j)\cos(\hat{\theta}^j) \end{bmatrix} \begin{bmatrix} u_x^j \\ u_y^j \end{bmatrix} \right) \tag{62}
$$

with

$$
\begin{aligned}
u_x^j &= {}^t\ddot{x}_{q_d}^j + \ddot{x}_t - \lambda_x^j \dot{e}_x^j - \kappa_x^j e_x^j - k_x^j s_x^j \\
u_y^j &= {}^t\ddot{y}_{q_d}^j + \ddot{y}_t - \lambda_y^j \dot{e}_y^j - \kappa_y^j e_y^j - k_y^j s_y^j \\
u_z^j &= {}^t\ddot{z}_{q_d}^j + \ddot{z}_t - \lambda_z^j \dot{e}_z^j - \kappa_z^j e_z^j - k_z^j s_z^j + g
\end{aligned} \tag{63}
$$

and where $k_x^j$, $\lambda_x^j$, $\kappa_x^j$, $k_y^j$, $\lambda_y^j$ and $\kappa_y^j$ are positive gains.

For the proposed control laws in Equations (58) and (62), it is necessary to know the accelerations of the target, which are estimated by deriving the estimated velocities of the target obtained from the EKF by using the following differentiator with the super-twisting algorithm [65] (see Figure 5):

$$
\begin{aligned}
\ddot{\hat{\mathbf{x}}}_\mathbf{t} &= -\mathbf{K_1} \left| \dot{\hat{\mathbf{x}}}_\mathbf{t} - \hat{\mathbf{v}}_\mathbf{t} \right|^{\frac{1}{2}} \operatorname{sign}(\dot{\hat{\mathbf{x}}}_\mathbf{t} - \hat{\mathbf{v}}_\mathbf{t}) + \hat{\omega}_t \\
\dot{\hat{\omega}}_t &= -\mathbf{K_2} \operatorname{sign}(\dot{\hat{\mathbf{x}}}_\mathbf{t} - \hat{\mathbf{v}}_\mathbf{t})
\end{aligned} \tag{64}
$$

where $\dot{\hat{\mathbf{x}}}_\mathbf{t}$ is the velocity of the target estimated by the differentiator, $\hat{\mathbf{v}}_\mathbf{t}$ is the velocity of the target estimated by the EKF, $\mathbf{K_1}$ and $\mathbf{K_2}$ are positive definite diagonal matrices of adequate dimensions.

To prove the stability of the flight formation subsystem, the following theorem is used:

**Theorem 1.** *Consider the dynamic in Equations (57) and (61) in closed loop with Equations (58) and (62); then, the origin $s_x^j = s_y^j = s_z^j = 0$ of Equations (56) and (60) is asymptotically stable.*

**Proof.** Consider the following Lyapunov candidate function:

$$
V_t = \frac{1}{2}\mathbf{s_t}^{j^\mathrm{T}} \mathbf{s_t} \tag{65}
$$

with $\mathbf{s_t}^j = \begin{bmatrix} s_x^j & s_y^j & s_z^j \end{bmatrix}^\mathrm{T}$. By deriving:

$$
\dot{V}_t = \mathbf{s_t}^{j^\mathrm{T}} \dot{\mathbf{s_t}}^j = \mathbf{s_t}^{j^\mathrm{T}} \begin{bmatrix} u_z^j \sin(\hat{\psi}^j) & u_z^j \cos(\hat{\psi}^j) & 0 \\ -u_z^j \cos(\hat{\psi}^j) & u_z^j \sin(\hat{\psi}^j) & 0 \\ 0 & 0 & \left(\frac{\mu^j}{m^j}\right)\cos(\hat{\phi}^j)\cos(\hat{\theta}^j) \end{bmatrix} \begin{bmatrix} \frac{\tan(\phi^j)}{\cos(\hat{\theta}^j)} \\ \tan(\theta^j) \\ u_q^j \end{bmatrix}
$$
$$
+ \mathbf{s_t}^{j^\mathrm{T}} \begin{bmatrix} \lambda_x^j \dot{e}_x^j + \kappa_x^j e_x^j - {}^t\ddot{x}_{q_d}^j - \ddot{x}_t \\ \lambda_y^j \dot{e}_y^j + \kappa_y^j e_y^j - {}^t\ddot{y}_{q_d}^j - \ddot{y}_t \\ -g + \lambda_z^j \dot{e}_z^j + \kappa_z^j e_z^j - {}^t\ddot{z}_{q_d}^j - \ddot{z}_t \end{bmatrix} \tag{66}
$$

Substituting Equations (58) and (62) in Equation (66), it is obtained:

$$
\dot{V}_t = -\mathbf{s_t}^{j^\mathrm{T}} \mathbf{K_t}^j \mathbf{s_t}^j \leq 0 \tag{67}
$$

with

$$\mathbf{K_t}^j = \begin{bmatrix} k_x^j & 0 & 0 \\ 0 & k_y^j & 0 \\ 0 & 0 & k_z^j \end{bmatrix} \tag{68}$$

Therefore, the origin $\mathbf{s_t}^j = \mathbf{0}$ of Equations (56) and (60) is asymptotically stable. □

### 5.2.2. Rotational Subsystem Control

For the rotational subsystem, with dynamics characterized by Equation (9), and control input $\tau^j$, it can be defined:

$$\mathbf{s_r}^j = \dot{\mathbf{e}}_\mathbf{r}^j + \lambda_r^j \mathbf{e_r}^j + \kappa_r^j \int_0^t \mathbf{e_r}^j dt \tag{69}$$

where the error signal is $\mathbf{e_r}^j = \sigma^j - \sigma_d^j$ and $\sigma_d^j$ is the desired value. Deriving Equation (69) and substituting in Equation (9), it is obtained:

$$\dot{\mathbf{s}}_\mathbf{r} = \mathbf{g}^j + \mathbf{B}^j \tau^j + \lambda_r^j \dot{\mathbf{e}}_\mathbf{r}^j + \kappa_r^j \mathbf{e_r}^j - \sigma_d^j \tag{70}$$

The following control law is proposed:

$$\tau^j = \mathbf{B}^{j^{-1}} \left( -\mathbf{g}^j + \sigma_d^j - \lambda_r^j \dot{\mathbf{e}}_\mathbf{r}^j - \kappa_r^j \mathbf{e_r}^j - \mathbf{K_r}^j \mathbf{s_r}^j \right) \tag{71}$$

where $\lambda_r^j$, $\kappa_r^j$, and $\mathbf{K_r}$ are positive definite diagonal matrices of adequate dimensions. It is straightforward to demonstrate that $\mathbf{B}^{j^{-1}}$ exists.

To prove the stability of the rotational subsystem, the following theorem is used:

**Theorem 2.** *Consider the dynamic in Equation (70) in closed loop with Equation (71); then the origin $\mathbf{s_r}^j = \mathbf{0}$ of Equation (69) is asymptotically stable.*

**Proof.** Consider the following Lyapunov candidate function:

$$V_r = \frac{1}{2} \mathbf{s_r}^{j^{\mathrm{T}}} \mathbf{s_r} \tag{72}$$

By deriving:

$$\dot{V}_r = \mathbf{s_r}^{j^{\mathrm{T}}} \dot{\mathbf{s}}_\mathbf{r}^j = \mathbf{s_r}^{j^{\mathrm{T}}} \left( \mathbf{g}^j + \mathbf{B}^j \tau^j + \lambda_r^j \dot{\mathbf{e}}_\mathbf{r}^j + \kappa_r^j \mathbf{e_r}^j - \sigma_d^j \right) \tag{73}$$

Substituting Equation (71) in Equation (73), it is obtained:

$$\dot{V}_r = -\mathbf{s_r}^{j^{\mathrm{T}}} \mathbf{K_r}^j \mathbf{s_r}^j \leq 0 \tag{74}$$

Therefore, the origin $\mathbf{s_r}^j = \mathbf{0}$ of Equation (69) is asymptotically stable. □

### 5.2.3. Closed Loop Stability

In order to prove the closed-loop stability of the whole system in Equation (54), the following theorem is used:

**Theorem 3.** *Given the system in Equation (54), in closed loop with the control laws in Equations (58) and (71). Moreover, if $\theta_d^j$ and $\phi_d^j$ are chosen according to Equation (62) as control laws for the longitudinal and lateral differential of the flight formation subsystem, then the origin $\mathbf{s_t}^j = \mathbf{s_r}^j = \mathbf{0}$ is asymptotically stable.*

**Proof.** Consider the following Lyapunov candidate function:

$$V_T = V_t + V_r \tag{75}$$

with derivative

$$\dot{V}_T = \dot{V}_t + \dot{V}_r = \mathbf{s_t}^{j\mathrm{T}}\dot{\mathbf{s}}_\mathbf{t}^j + \mathbf{s_r}^{j\mathrm{T}}\dot{\mathbf{s}}_\mathbf{r}^j \tag{76}$$

from Equations (67) and (74), it can be seen that

$$\dot{V}_T = -\mathbf{s_t}^{j\mathrm{T}}\mathbf{K_t}\mathbf{s_t} - \mathbf{s_r}^{j\mathrm{T}}\mathbf{K_r}\mathbf{s_r} \leq 0 \tag{77}$$

Therefore, the origin $\mathbf{s_t}^j = \mathbf{s_r}^j = \mathbf{0}$ of the closed loop system is Globally Asymptotically Stable (GAS). □

## 6. Simulation Results

### 6.1. Simulation Setup

In this section, the proposed cooperative visual-SLAM system for UAV-based target tracking is validated through computer simulations. A Matlab$^{\circledR}$ implementation of the proposed scheme was used for this purpose. To this aim, a simulation environment has been developed. The environment is composed of $3D$ landmarks, randomly distributed over the ground (See Figure 6). To execute the tests, two quadcopters (Quad 1 and Quad 2), equipped with the set of sensors required by the proposed method are simulated. In simulations, it is assumed that there exist enough landmarks in the environment that allow for being observed in common by the cameras of the UAVs, at least a subset of them.

The measurements from the sensors are emulated to be taken with a frequency of 10 Hz. To emulate the system uncertainty, the following Gaussian noise is added to measurements: Gaussian noise with $\sigma_c = 4$ pixels is added to the measurements given by the cameras. Gaussian noise with $\sigma_a = 25$ cm is added to the measurements of altitude differential between two quadcopters obtained by the altimeters. It is important to note that the noise considered for emulating monocular measurements is bigger than the typical magnitude of the noise of real monocular measurement. In this way, the errors in camera orientation are considered caused by the gimbal, in addition to the imperfection of the sensor.

In simulations, the target was moved along a predefined trajectory (see Figure 6).

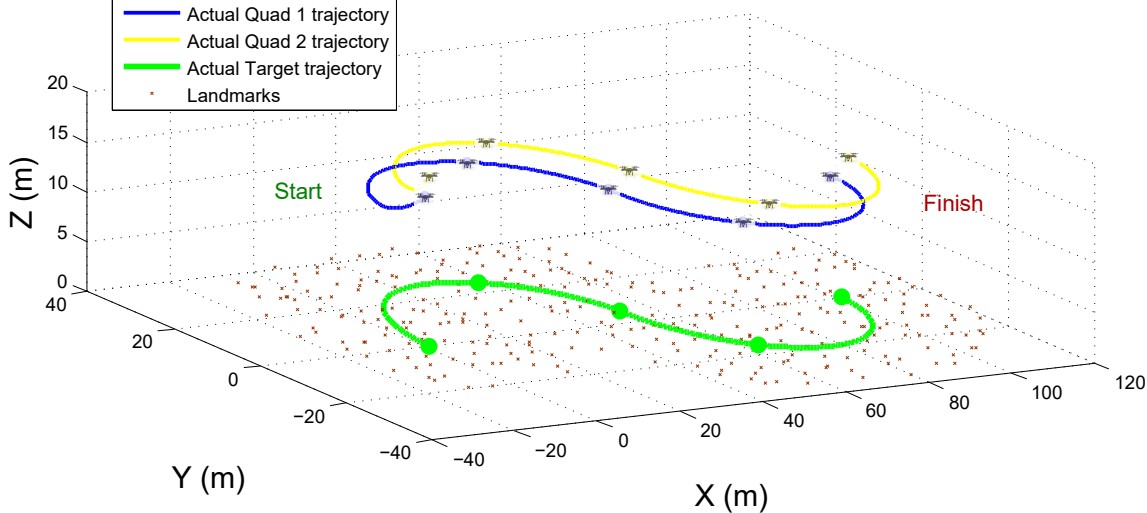

**Figure 6.** Actual trajectories followed by the target, Quad 1 and Quad 2.

The parameter values used for the quadcopters and the cameras are shown in Table 2. The quadcopters' parameters are like those presented in [66]. The camera parameters are like those presented in [17].

**Table 2.** Quadcopter and camera parameters.

| Parameter | Value | Parameter | Value |
|-----------|-------|-----------|-------|
| $m^j$ | 0.468 | $l^j$ | 0.225 |
| $\mu^j$ | $2.98 \times 10^{-6}$ | $g$ | 9.81 |
| $J_p^j$ | $3.357 \times 10^{-5}$ | $f^j/d_u^j$ | 262.92 |
| $d^j$ | $1.14 \times 10^{-7}$ | $f^j/d_v^j$ | 261.66 |
| $I_x^j, I_y^j$ | $4.856 \times 10^{-3}$ | $c_u^j$ | 359.51 |
| $I_z^j$ | $8.801 \times 10^{-3}$ | $c_v^j$ | 239.50 |

Figure 6 shows the trajectories of the three elements composing the flight formation. To highlight them, the position of the elements of the formation is indicated at different instants of time.

*6.2. Convergence and Stability Tests*

In a series of tests, the convergence and stability of the proposed cooperative visual-SLAM system, in open-loop, were evaluated under the three different observability conditions described in Section 3.2. In this case, it is assumed that there is a control system able to maintain the aerial robots flying in formation with respect to the target.

Two different kinds of tests were performed:

(a) For the first test, the robustness of the cooperative visual-SLAM system with respect to errors in the initialization of map features is tested. In this case, the initial conditions of ${}^t\hat{\mathbf{x}}_c^1$, ${}^t\hat{\mathbf{x}}_c^2$, ${}^t\hat{\mathbf{v}}_c^1$, ${}^t\hat{\mathbf{v}}_c^2$ and $\hat{\mathbf{v}}_t$ are assumed to be known exactly, but each $i$ map feature is forced to be initialized into the system state with a determined error position. Three different conditions of initial error are considered: $\sigma_i = \{0.5, 1.5, 2.25\}$ meters.

(b) For the second test, additional to the map features initial errors, the initial conditions of ${}^t\hat{\mathbf{x}}_c^1$, ${}^t\hat{\mathbf{x}}_c^2$, ${}^t\hat{\mathbf{v}}_c^1$, ${}^t\hat{\mathbf{v}}_c^2$, and $\hat{\mathbf{v}}_t$ are not known exactly and have a considerable error with respect to the real values. In this case, the robustness of the proposed estimation system is evaluated in function to the errors in the initial conditions of all the state variables.

Figure 7 and Figure 8 show respectively the results for tests (a) and (b). In this case, the estimated relative position of the Quad 1 with respect to the target is plotted for each reference axis (row plots). Note that, for the sake of clarity, only the estimated values for Quad 1 are presented. The results for Quad 2 are quite similar to those presented for Quad 1. In these figures, each column of plots shows the results obtained from an observability case.

In both figures, it can be observed that both the observability property and the initial conditions play a preponderant role in the convergence and stability of the EKF-SLAM. For several applications, at least the initial position of the UAVs is assumed to be known. However, in SLAM, the position of the map features must be estimated online. The above outcome confirms the importance of using good feature initialization techniques in visual-SLAM. Of course, as it can be expected, the performance of the EKF-SLAM is considerably better when the system is observable. Note that, for the observability case 3, and despite a noticeable error drift, the filter still exhibits certain stability for the worst case of features initialization, when the initial conditions of the UAVs are known.

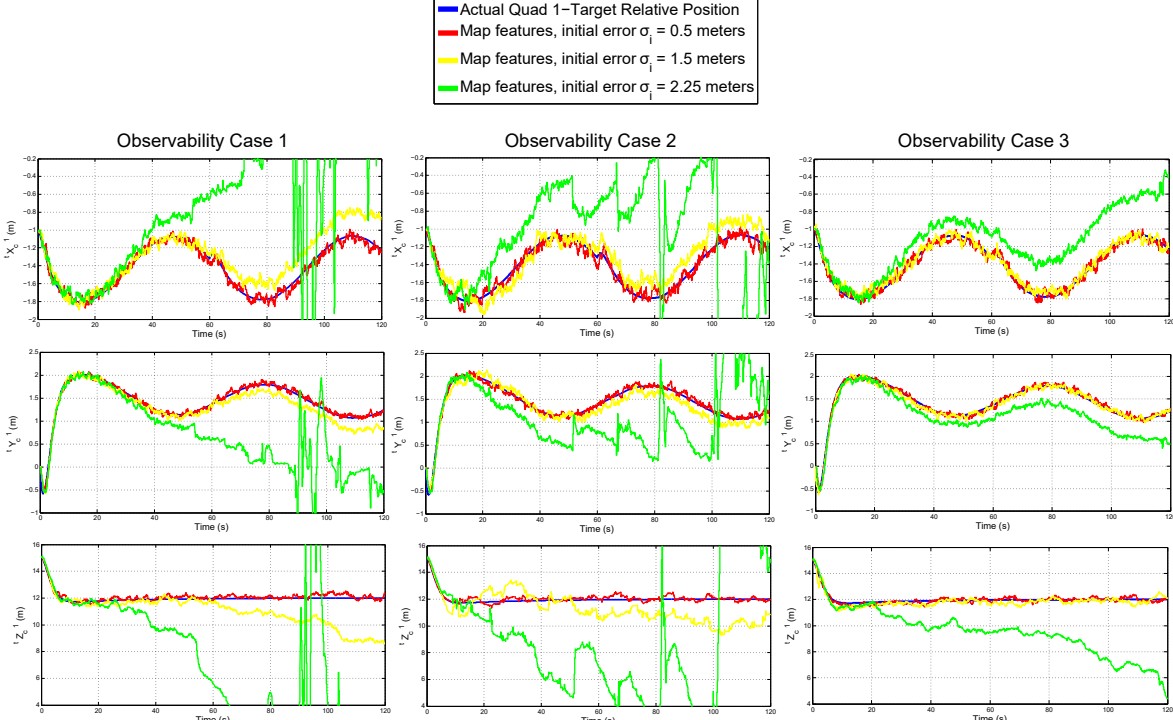

**Figure 7.** The estimated relative position of the Quad 1 with respect to the target obtained by the Test (**a**).

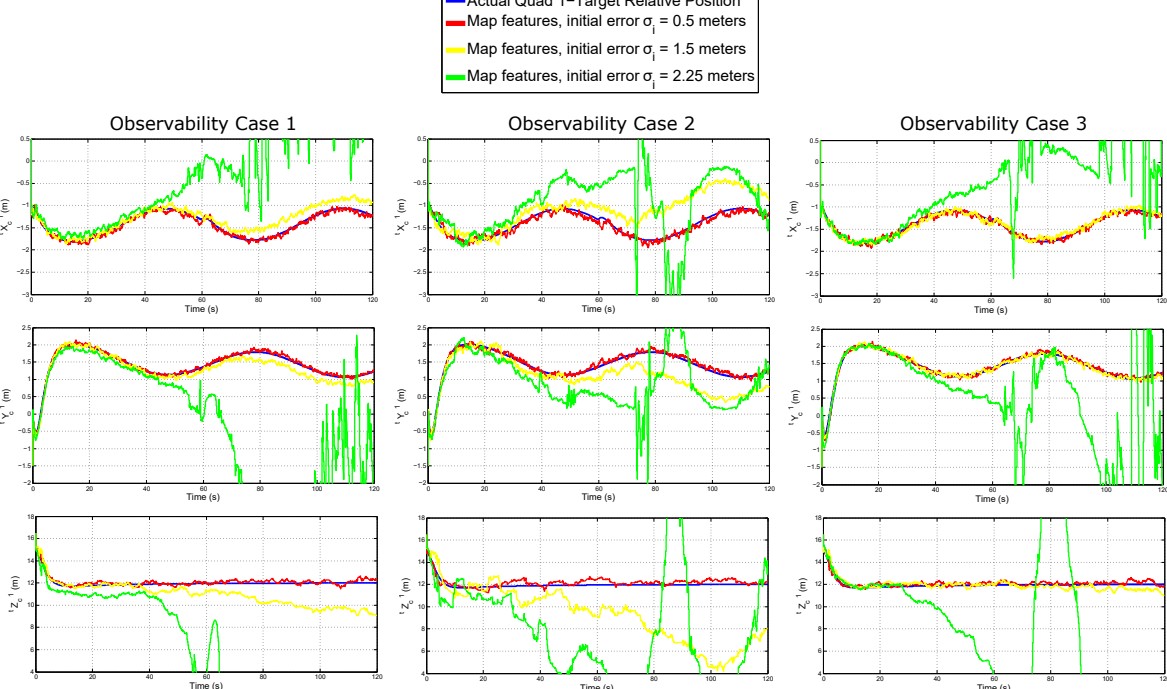

**Figure 8.** The estimated relative position of the Quad 1 with respect to the target obtained by the Test (**b**).

### 6.3. Comparative Study

Using the same simulation setup, for this series of tests, the performance of the proposed system, for estimating the relative position of the UAVs with respect to the moving target, was evaluated through a comparative study. Table 3 summarizes the characteristics of five UAV-based target-tracking methods used in this comparative study.

**Table 3.** Characteristics of the methods used in the comparative study.

| Method | System Configuration | Parametrization | Visual Measurements | Other Measurements |
| --- | --- | --- | --- | --- |
| Proposed | Multi-UAV-SLAM | Target-centric | Monocular | Altitude differential |
| (1) | Single-UAV-SLAM | Target-centric | Monocular | - |
| (2) | Multi-UAV-SLAM | Robot-centric | Monocular | Altitude differential |
| (3) | Single-UAV | Robot-centric | Stereo-vision | - |
| (4) | Multi-UAV | Robot-centric | Pseudo-stereo | GPS |
| (5) | Single-UAV | Robot-centric | Monocular | Range |

There are some remarks about the methods used in the comparison. Method (1) represents a purely standard monocular-SLAM approach. Features initialization is based on [67]. Because the metric scale cannot be retrieved using only monocular vision, it is assumed that the positions of landmarks seen in the first frame are perfectly known. Method (2) is similar to the proposed previous method, but the system is robot-centric parametrized, instead of target-centric. Method (3) represents an standard visual-stereo tracking approach. In this case, the moving target position is directly obtained by a stereo system, with a baseline of 15 cm. The method (4) is based on the approach presented in [68]. In this case, UAVs are equipped with GPS, and the moving target position is estimated through the pseudo-stereo vision system composed of the monocular cameras of each UAV. Gaussian noise with $\sigma_g = 20$ cm is added to GPS measurements. Method (5) estimates the target position using monocular and range measurements to the target. In this case, it is assumed some cooperative-target scheme for obtaining range measurements. Technology for obtaining such kind of range measurements is presented in [69]. Gaussian noise with $\sigma_r = 20$ cm is added to the range measurements. Note that methods (3), (4), and (5) are not SLAM methods. Methods (3) and (5) provide only the relative estimation of the target position with respect to the UAV. Method (4), due to the GPS, provides also global position estimations of the target and UAVs.

In case of SLAM methods and to carry out a more realistic evaluation, the data association problem was also accounted for. To emulate the failures of the visual data association process, 5% of the total number of visual correspondences are forced to be outliers in a random manner. Table 4 shows the number of outliers introduced into the simulation due to the data association problem. Outliers for the visual data association in each camera as well as outliers for the cooperative visual data association are considered.

**Table 4.** Number of outliers introduced into the SLAM methods.

| Method | Visual Outliers (Quad 1) | Visual Outliers (Quad 2) | Visual Outliers (Cooperative) |
| --- | --- | --- | --- |
| Proposed | 3798 | 3798 | 2338 |
| (1) | 3820 | - | - |
| (2) | 3798 | 3798 | 2338 |

Figure 9 shows the results obtained from each method for estimating the relative position of Quad 1 with respect to the target. For the sake of clarity, the results are shown in two columns of plots. Each row of plots represents a reference axis.

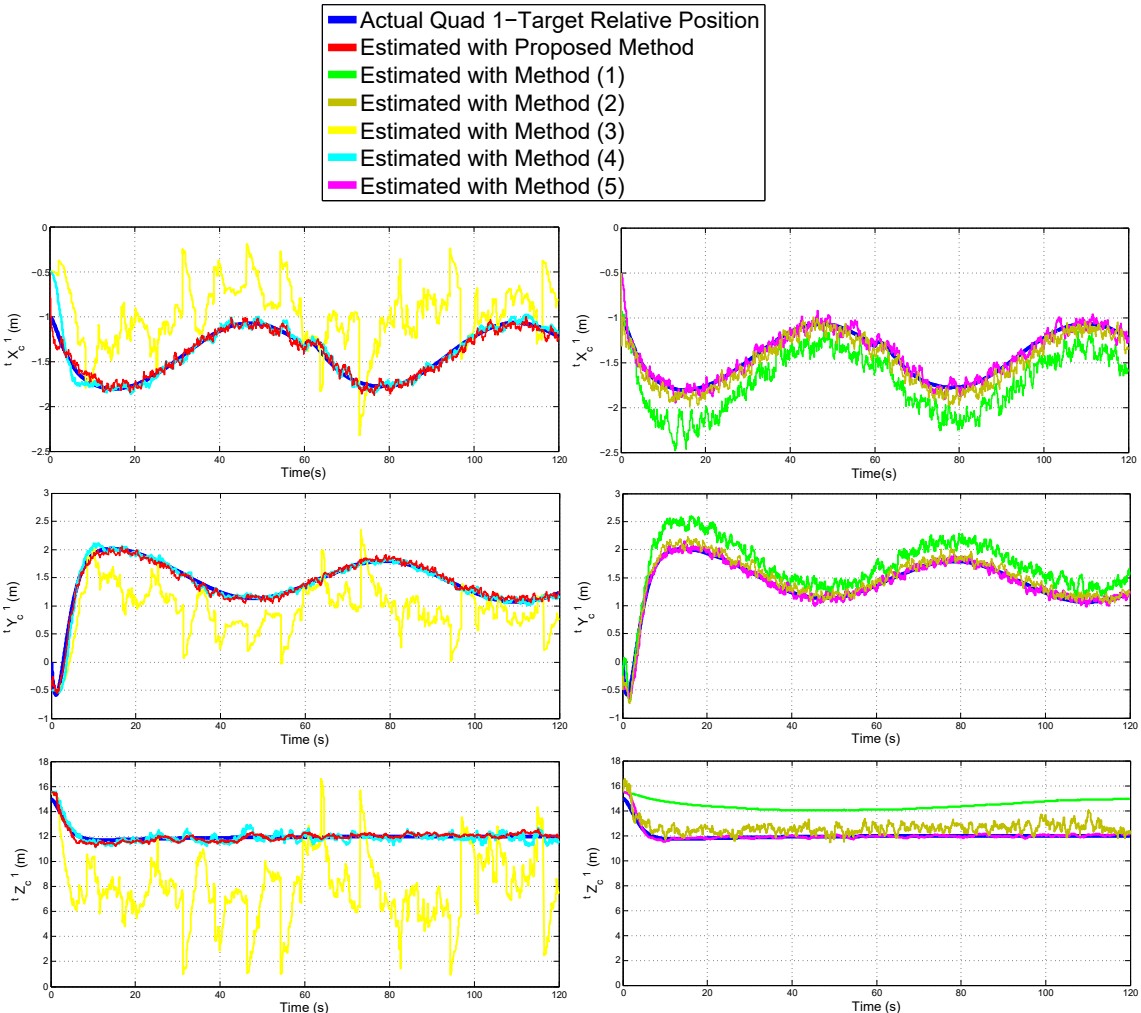

**Figure 9.** The estimated relative position of the Quad 1 with respect to the target obtained from the six systems.

Table 5 summarizes the Mean Squared Error (MSE) for the estimated relative position of Quad 1 with respect to the target in the three axes.

**Table 5.** Total Mean Squared Error of the estimated relative position of Quad 1 with respect to the target.

| Method | MSEX (m) | MSEY (m) | MSEZ (m) |
|--------|----------|----------|----------|
| Proposed | 0.0029 | 0.0032 | 0.0512 |
| (1) | 0.0975 | 0.1053 | 6.0702 |
| (2) | 0.0083 | 0.0090 | 0.5238 |
| (3) | 0.0069 | 0.0048 | 0.0937 |
| (4) | 0.2829 | 0.3539 | 23.4294 |
| (5) | 0.0045 | 0.0036 | 0.0230 |

Regarding the performance of SLAM methods for estimating the features map, Table 6 shows the total (sum of all) Mean Squared Errors for the estimated position of landmarks, while Table 7 shows the total Mean Squared Errors for the initial estimated position of the landmarks.

**Table 6.** Total Mean Squared Error of the estimated position of landmarks.

| Method | MSEX (m) | MSEY (m) | MSEZ (m) |
|---|---|---|---|
| Proposed | 0.0969 | 0.0687 | 0.1991 |
| (1) | 6.1459 | 6.2934 | 5.9953 |
| (2) | 0.4231 | 0.3471 | 0.7477 |

**Table 7.** Total Mean Squared Error of the estimated initial position of landmarks.

| Method | MSEX (m) | MSEY (m) | MSEZ (m) |
|---|---|---|---|
| Proposed | 0.9993 | 0.8787 | 1.2064 |
| (1) | 4.3668 | 4.4917 | 3.8829 |
| (2) | 1.4241 | 1.2655 | 1.7639 |

According to the above results, the relative position of the UAV with respect to the moving target was recovered fairly well with the following methods: authors' proposed Target-Centric Multi-UAV SLAM approach, (2) Robot-Centric Multi-UAV SLAM, (4) Multi-UAV Pseudo-stereo, and (5) Single-UAV monocular-range.

It is important to note that method (4) relies on GPS and thus it is not suitable for GPS-denied environments. Method (5) relies on range measurements to the target which can be often difficult to obtain, or it requires some cooperative target scheme which is not even possible to accomplish for several kinds of applications. Regarding the other approaches, method (1) does not have direct three-dimensional information of the target (only monocular measurement) and thus the estimation is not good. In the case of method (3), it is well known that, with a fixed stereo system, the quality of measurements considerably degenerates as the distance with respect to the target increases.

Thus far, the above results suggest that both the proposed approach and the method (2) (both SLAM systems) exhibit good performance in challenging conditions: (i) GPS-denied environments, (ii) non-cooperative target, and (iii) long distance to the target. The only difference between these two methods is the parametrization of the system (Target-centric vs. Robot-centric). To investigate if there is an advantage in the use of one parametrization over the other, another test was carried out.

In this case, using the same simulation setup, the robustness of the methods was tested against the loss of visual contact to the moving target by one UAV during some period. Figure 10 shows the estimated relative positions of Quad 1 with respect to the target obtained with both systems. Note that, during the seconds 45 to 75 of simulation, there are no monocular measurements of the target done by Quad 1. Only the estimated values of Quad 1 are presented, but the results of Quad 2 are very similar.

Given the above results, it can be seen that, contrary to the method (2), the proposed system is robust to the loss of visual contact of the target by one of the Quads. This is a clear advantage of the proposed Target-centric parameterization.

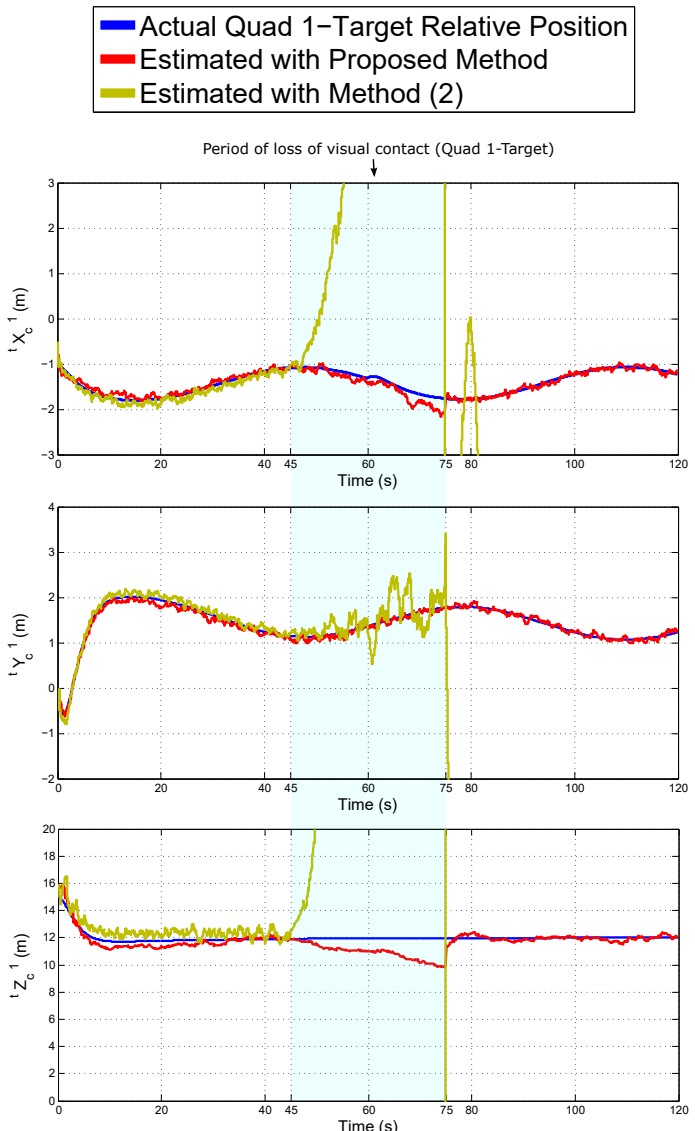

**Figure 10.** Estimated relative position of Quad 1 with respect to the target with a period of loss of visual contact of the target by Quad 1.

*6.4. Control Simulations' Results*

A set of simulations was also carried out to test the whole proposed system in an estimation-control closed-loop manner. In this case, to maintain a stable flight formation with respect to the moving target, the estimates obtained from the proposed visual-based estimation system are used as feedback to the control scheme described in Section 5).

In this test, the values of vectors ${}^{t}\mathbf{x}_{\mathbf{q}_d}^{i}$ that define the flight formation are:

For Quad 1: ${}^{t}\mathbf{x}_{\mathbf{q}_d}^{1} = \left[ \begin{array}{ccc} -1.4142 - 0.35\sin(t \cdot 0.1), & 1.4142 + 0.35\sin(t \cdot 0.1), & 12 \end{array} \right]^{T}$.

For Quad 2: ${}^{t}\mathbf{x}_{\mathbf{q}_d}^{2} = \left[ \begin{array}{ccc} 2 + 0.5\sin(t \cdot 0.1), & 0, & 14 \end{array} \right]^{T}$.

In addition, in this test, a period of visual contact loss of target by Quad 1 is simulated (seconds 35 to 50). A period of a visual contact loss of target by both Quads is simulated (seconds 90 to 95) as well.

Figure 11 shows the evolution of the error with respect to the desired values ${}^{t}\mathbf{x}_{\mathbf{q}_d}^{i}$. In all the cases, note that the errors are bounded after an initial transient period.

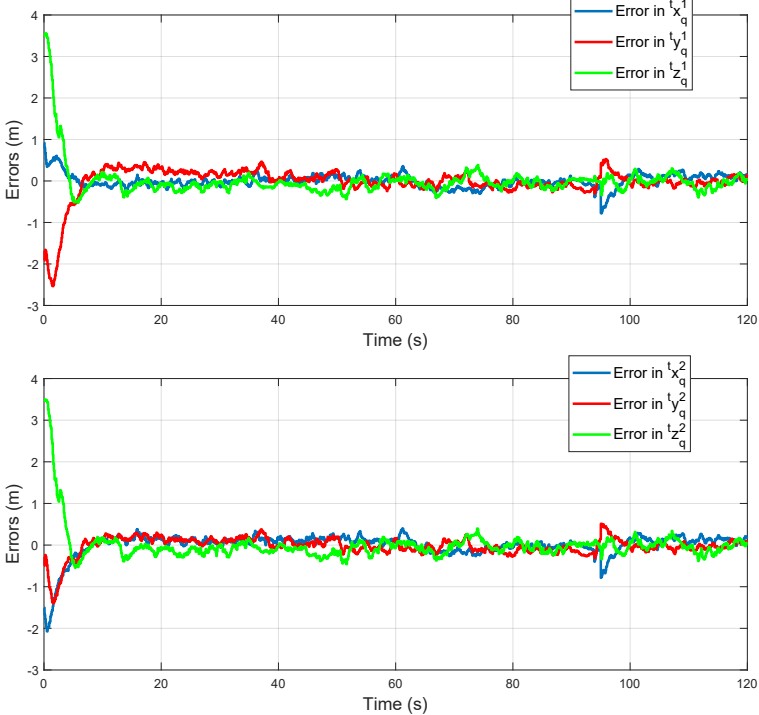

**Figure 11.** Errors in $\mathbf{^t x_{q}}_d^{\,i}$ during the flight trajectories.

Figure 12 shows the real and estimated relative position of Quad 1 (left column) and Quad 2 (right column) with respect to the target. Table 8 summarizes the Mean Squared Error (MSE), in each axis, for the estimated relative position of Quad 1 and Quad 2 with respect to the target.

**Table 8.** Total Mean Squared Error in the estimated relative position of Quad 1 and Quad 2 with respect to the target.

|        | MSEX (m) | MSEY (m) | MSEZ (m) |
|--------|----------|----------|----------|
| Quad 1 | 0.0137   | 0.0080   | 0.0738   |
| Quad 2 | 0.0134   | 0.0068   | 0.0848   |

According to the above results, it can be seen that the control system was able to maintain a stable flight formation along with all the trajectory with respect to the target, using the proposed visual-based estimation system as feedback.

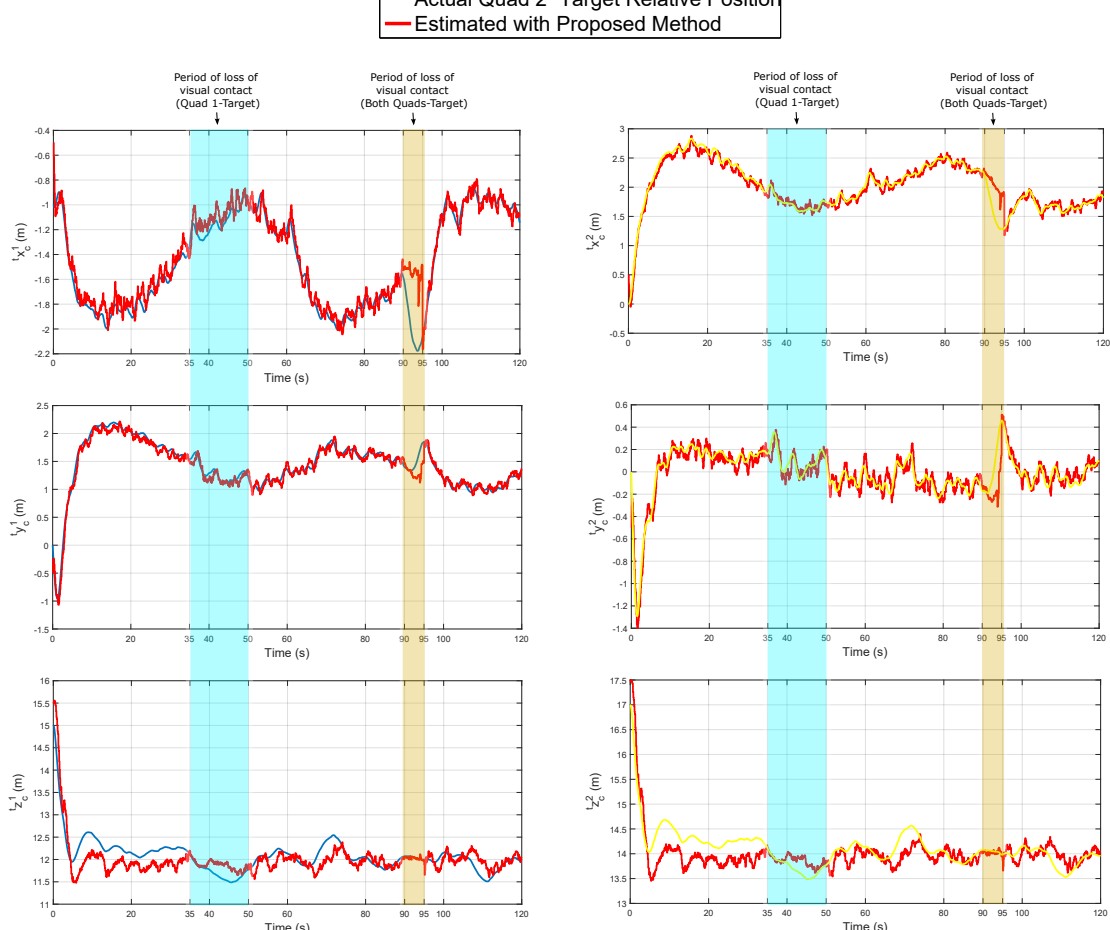

**Figure 12.** The estimated relative position of Quad 1 and Quad 2 with respect to the target.

## 7. Conclusions

This research presents a cooperative visual-based SLAM system that allows a team of aerial robots autonomously following a non-cooperative target moving freely in a GPS-denied environment. The relative position of each UAV with respect to the target is estimated by fusing monocular measurements of the target and landmarks using the standard EKF-based SLAM methodology.

The observability property of the system was investigated using an extensive nonlinear observability analysis. In this case, new theoretical results were presented and the observability conditions were derived. To improve the observability properties of the system, and thus its performance, the use of a Target-centric SLAM configuration is proposed. In addition, measurements of altitude differential between pairs of UAVs, obtained from altimeters, are integrated into the system for the same purpose. Additionally to the proposed estimation system, a control scheme was proposed allowing the control of UAVs flight formation with respect to the moving target. The stability of control laws is proved using the Lyapunov theory. An extensive set of computer simulations was performed in order to validate the proposed scheme. According to the simulation results, the proposed system shows an excellent and robust performance to estimate the relative position of each UAV with respect to target. The results also suggest that the proposed approach has better performance when it is compared with other related methods. Moreover, with the proposed control laws, the contributed SLAM system demonstrates good performance in closed-loop. On the other hand, although computer simulations are useful for evaluating the full statistical consistency of the methods, they can still neglect important practical issues that appear when the methods are applied in real scenarios. Accordingly,

it is important to note that future work will be focused on developing experiments with real data in order to fully evaluate the applicability of the proposed approach.

**Author Contributions:** Conceptualization, R.M. and A.G.; methodology, S.U and R.M.; software, J.-C.T. and S.U.; validation, J.-C.T., S.U., and A.G.; investigation, S.U. and R.M.; resources, J.-C.T. and S.U.; writing—original draft preparation, J.-C.T. and R.M.; writing—review and editing, R.M. and A.G.; supervision, R.M. and A.G.; funding acquisition, A.G. All authors have read and agreed to the published version of the manuscript.

**Funding:** This research has been funded by Project DPI2016-78957-R, Spanish Ministry of Economy, Industry and Competitiveness.

**Conflicts of Interest:** The authors declare no conflict of interest.

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
