# Peer review of "Cooperative Visual-SLAM System for UAV-Based Target Tracking in GPS-Denied Environments: A Target-Centric Approach"

_electronics, doi:10.3390/electronics9050813_

Round 1

Reviewer 1 Report

            The paper “Cooperative Visual-SLAM System for UAV-based Target Tracking in GPS-denied Environments: A Target-centric approach” focuses on a very important topic of target tracking by UAV systems in areas where GNSS signals are either obscured or intentionally denied. This is a very important topic that emerged in the last few years and I believe that the authors do a good job in both exploring the background of this problem, discuss the strengths and weakness of currently employed methods, and present a novel solution based on visual assessment of target and subsequent assessment of its geographical position through relative localization and mapping. Innovatively, the authors propose a target-centric approach to the location problem, which I found to be an interesting and novel solution. While I enjoyed the paper and I believe that it can be accepted after a round of revisions, there are a number of questions that should be addressed.

            Specifically, there are three broad questions that I believe need to be addressed in terms of the work’s limitation, they all stem from the authors focusing on a target that can move freely in 3D space (Line 104):

  1. It is not clear how data from the UAV swarm engaging a target will be processed and relayed back to the decision center. In essence, if the data is processed on board the UAV and calculations/decisions are made autonomously, then what are the mass limitations of such a system and is this something that needs to be mounted on a separate heavy-lift UAV. If the datasets are relayed and processed elsewhere, then how would this be achieved, assuming that in a GNSS-denied environment other systems of wireless communications may also be jammed by radioelectronic warfare units?
  2. The study assumes that the target consistently stays below the horizonal line of sight of the system, is that a realistic expectation, given that the target can be a lighter, higher-flying UAV?
  3. Finally, if the target is in fact launched with malicious intent, the intruding vehicle can (and will) come in at either extremely low altitude, or an extremely high altitude, using cloud, smoke, or fog cover to obscure line of sight. The proposed system is built around obtaining and holding a visual line of sight to the target from multiple UAVs. How would the system behave if that line of sight is broken or if it is obscured at any given moment?
  4. Finally, would this system be effective during night-time hours? This is a critical consideration in terms of judging its effectiveness against LiDAR-type systems.

In all, despite some of my concerns, I enjoyed reading this manuscript and I look forward to seeing a revised version, as I believe the authors present on a very important topic.

Author Response

Dear reviewer, 

find attached in a file our response to your valuable comments, 

the authors

Reviewer 2 Report

This paper proposes a target-centric cooperative visual-based SLAM system without GPS signal aiding. This system integrates the measurements of altitude differential between pairs of UAVs for addressing the problem of estimating the relative position of the aerial robots with respect to the target. That seems to have good operability, but I believe that some revisions are required before publication in the journal.

Which reference system are the two relative position variables shown in lines 156 to 157 with respect to?

There are syntax errors in the line 493 and the vector in (A1) has incorrect descriptions, please check them.

The fourth part of the paper seems to concentrate on describing EKF and lack a detailed expression on the SLAM method.

Author Response

(The authors gave the same response as above.)

Reviewer 3 Report

An interesting approach to UAV navigation in a non-GPS environment. This is very important in the era of jamming and spoofing of GPS signal. Autonomous vehicles are developing rapidly and are undoubtedly a technology of the future. Article is well written and organised.

Some remarcs:
1) Fig.4. usually in navigation, geodesy and related sciences the x-axis is oriented to the north because of the clockwise measurement of courses, azimuth and bearing from the north.
2) Variables in the formulas should be defined. On the following pages of the article there are new markings of variables that are not named, which makes it very difficult to read the article.
3) Kalman filtering is well known and rather old method. Promising method are Artificial Neural Networks also used for target tracking and similar tasks. Analysis of neural approach would be advisable. The following publications would be advisable for analysis:
* Kazimierski, W., Lubczonek, J. Verification of marine multiple model neural tracking filter for the needs of shore radar stations. Book Editor(s): Kulpa, K.13th International Radar Symposium (IRS). Book Series: International Radar Symposium Proceedings pp. 554-559 Warsaw (2012).
*Kazimierski, W., Zaniewicz G., Stateczny, A.: Verification of multiple model neural tracking filter with ship's radar. Book Editor(s): Kulpa, K.13th International Radar Symposium (IRS). Book Series: International Radar Symposium Proceedings, pp. 549-553, Warsaw (2012).
*Stateczny, A.: Neural manoeuvre detection of the tracked target in ARPA systems. Book Editor(s): Katebi, R. Control Applications in Marine Systems 2001 (CAMS 2001). Book Series: IFAC Proceedings Series, pp. 209-214, Glasgow (2002).

Author Response

(The authors gave the same response as above.)
